# Dynamics of neural scaling laws in random feature regression with powerlaw-distributed kernel eigenvalues

**Jakob Kramp** [* 1 2]   **Javed Lindner** [* 1 2]   **Moritz Helias** [* 1 3]

## Abstract

Training large neural networks exposes neural scaling laws for the generalization error, which points to a universal behavior across network architectures of learning in high dimensions. It was also shown that this effect persists in the limit of highly overparametrized networks as well as the Neural network Gaussian process limit. We here develop a principled understanding of the typical behavior of generalization in Neural Network Gaussian process regression dynamics. We derive a dynamical mean-field theory that captures the typical case learning dynamics: This allows us to unify multiple existing regimes of learning studied in the current literature, namely Bayesian inference on Gaussian processes, gradient flow with or without weight-decay, and stochastic Langevin training dynamics. Employing tools from statistical physics, the unified framework we derive in either of these cases yields an effective description of the high-dimensional microscopic behavior of networks dynamics in terms of lower dimensional order parameters. We show that collective training dynamics may be separated into the dynamics of $N$ independent eigenmodes, whose evolution equations are only coupled through collective response functions and a common statistics of an effective, independent noise. Our approach allows us to quantitatively explain the dynamics of the generalization error by linking spectral and dynamical properties of learning on data with power law spectra, including phenomena such as neural scaling laws and the effect of early stopping.

## 1. Introduction

Natural data, such as language and image data, ubiquitously shows power law distributions in their principal component spectra (Schaaf & Hateren; Koch et al.; Bulanadi & Paruch). This property persists even if the data is mapped into a feature space. Training large neural networks on such data exposes the phenomenon of neural scaling laws for the generalization error, which points to a universal behavior of learning in high dimensions. Such behavior is observed across neural architectures (Kaplan et al., 2020). The relevance of this universal behavior also has practical applications, as it provides a theory based approach to use smaller and computationally more efficient models to gauge the behavior of bigger models without running costly training. It was also shown that this universality persists in the limit of large numbers of neurons, which reduces networks to Gaussian process regression (Jacot et al., 2018; Lee et al., 2018). Gaussian process regression, and thus also linear regression, offers an analytically tractable setting that captures key qualitative behaviors of real-world overparameterized networks, including scaling laws and the fabled double descent phenomenon (Nakkiran et al., 2021). We thus focus our work on a principled understanding of the typical behavior of generalization in linear regression dynamics.

To this end, we derive a dynamical mean-field theory that captures the typical learning dynamics in the limit of large numbers of neurons and training samples. This approach allows us to unify multiple existing approaches to the study of learning found in the current literature, namely Bayesian inference on Gaussian processes, gradient flow with or without weight-decay, and stochastic Langevin training dynamics. We arrive at this versatile theory by employing statistical field theory of disordered systems. As in statistical physics, the high-dimensional microscopic behavior of the network's dynamics can be effectively described by a set of interpretable lower dimensional order parameters. The theory shows that the collective training dynamics can be separated into the dynamics of $N$ independent eigenmodes, whose evolution equations are only coupled through a collective response function and the jointly determined statistics of an effective noise. This approach enables us to link the success

*Equal contribution [1] Institute for Advanced Simulation (IAS-6), Computational and Systems Neuroscience, Jülich Research Centre, Jülich, Germany [2] RWTH Aachen University, Aachen, Germany [3] Department of Physics, RWTH Aachen University, Aachen, Germany. Correspondence to: Jakob Kramp <jakob.kramp@rwth-aachen.de>.

*Proceedings of the 43rd International Conference on Machine Learning*, Seoul, South Korea. PMLR 306, 2026. Copyright 2026 by the author(s).

of machine learning heuristics like "early-stopping" (see e.g. (Advani et al., 2020)) to the spectral and dynamical properties of learning on power law distributed data. Specifically, we

- derive an effective mean-field theory that quantitatively explains the dynamics of the generalization error, including phenomena such as neural scaling laws and the effect of early stopping,

- explain how the analytically found, disorder induced, effective self-coupling slows down the learning dynamics,

- and obtain analytical results that relate the power-law exponent of the feature kernel, regularization, and early stopping time to obtain a minimal generalization gap.

## 2. Related works

Past work has focused on different aspects of learning that we unify here: Krogh & Hertz (1990; 1992) and Krogh (1992) were the first to consider the generalization properties of linear regression, its learning dynamics in a stochastic setup and the influence of data and network properties on learning dynamics as well as dynamics close to the interpolation threshold. The setup is close to ours, but differs by them not considering kernel regression with power-law distributed eigenvalues. Dunmur and Wallace (1993) studied the effects of noisy data on the performance of linear systems in a stochastic setup at infinite training time $t \to \infty$. Further approaches include computing the asymptotic limit at $t \to \infty$ utilizing Green's function approaches (Sollich, 1994), where Halkjær & Winther (1996) also considered the effects of modes of the input covariance on the convergence of gradient descent.

More recent literature is structured into static and dynamic approaches in the following.

Among the static approaches, which is the limit of infinite training time, our work is methodologically closest related to Canatar et al. (2021) and Bordelon et al., who consider feature regression in a reproducing kernel Hilbert space with power-law distributed kernel eigenvalues. While technically the authors introduce stochasticity (finite temperature), they do not exploit the link to Bayesian inference which we expose here. Cui, Loureiro, Krzakala, and Zdeborová (2022) extend Bordelon et al. by adding label noise in their setup. Spigler, Geiger, and Wyart compare student-teacher kernel regression to soft SVM classification and derive generalization error exponents from smoothness and dimensionality assumptions. Maloney, Roberts, and Sully (2022) investigated neural scaling laws in linear regression with power-law distributed eigenvalue spectra and L2-regularization. Defilippis, Loureiro, and Misiakiewicz study static random

feature regression without specific scaling assumptions on sample or system size. Atanasov, Zavatone-Veth, and Pehlevan (2024) derive deterministic equivalents for static random feature regression via the S-transform, compactly charting scaling law exponents across many settings.

On the side of the dynamical approaches, Advani et al. (2020) studied the dynamics of deterministic gradient flow in linear regression with L2 regularization; compared to our work, their work lacks temporal stochasticity and power-law distributed eigenvalue spectra; instead their data is Gaussian i.i.d. Bordelon, Atanasov, and Pehlevan (2024) studied learning dynamics as gradient flow with power law data in a type of linear random feature models, which showed how effects of feature learning may affect neural scaling laws, while considering systems in a gradient flow regime. Our setup is most closely related to theirs, however, they do not cover the effects of L2 regularization and in addition our work requires fewer and more interpretable order parameters. While having access to the dynamics, the authors do not investigate the response functions and the resulting slowdown of the training dynamics further, but rather focus on the time dynamics of the loss. Paquette, Paquette, Xiao, and Pennington study the compute-optimal Pareto frontier in random feature regression trained by SGD with finite batch size and power-law features, deriving a deterministic Volterra equation via random matrix theory and identify multiple scaling phases. Like Bordelon et al. (2024), they use a down-projected student-teacher setup. We instead study Langevin gradient flow with identical student and teacher features, which, beyond feature regression, naturally connects to the Bayesian posterior.

## 3. Setup

We consider a teacher-student setup (Krogh & Hertz, 1992) with

$$f_\mu = w^\top \psi_\mu \, , \qquad (1)$$

$$y_\mu = \overline{w}^\top \psi_\mu \, , \qquad (2)$$

where $\psi(x_\mu) =: \psi_\mu \in \mathbb{R}^N$ are considered the features of the data $x_\mu$ and $\mu$ indexes the training points. The labels $y_\mu$ are generated by the teacher weights $\overline{w} \in \mathbb{R}^N$ while $w \in \mathbb{R}^N$ are the student weights our model aims to learn. While we consider a noiseless teacher here, label noise can easily be included in the theory as demonstrated in Section H. The train set for this regression task is given by $\mathcal{D} = \{(x_\mu, y_\mu)\}_{\mu=1,\dots,P}$ and $(x_*, y_*) \notin \mathcal{D}$ is a test point. We assume $x_\mu, x_* \overset{\text{i.i.d.}}{\sim} p(x)$ to be drawn from the same distribution. We solve the regression task on the teacher-student setup with an L2-regularized squared loss

$$H(w, \mathcal{D}) := \frac{1}{2} \sum_{\mu=1}^{P} \left( y_\mu - f_\mu(w) \right)^2 + \frac{1}{2g\beta} \sum_{i=1}^{N} w_i^2 \quad . \quad (3)$$

Instead of considering gradient flow, which would yield a single maximum a posteriori point estimate for the weights and the network output, we consider a setting where the loss is minimized under stochastic Langevin gradient descent, which has been proven to serve as a good approximation to more realistic training scenarios (see e.g. (Naveh et al., 2020)). The update equation for the weights reads

$$\frac{\partial}{\partial t} w_i(t) = -\partial_{w_i} H\big(w(t), \mathcal{D}\big) + \zeta_i(t) , \quad (4)$$

$$\langle \zeta_i(t)\zeta_j(s)\rangle_\zeta = 2/\beta \, \delta_{ij} \, \delta(t-s) \quad . \quad (5)$$

This setup for regression is general in the sense that it covers three aspects: Firstly, $\beta \to \infty$ yields gradient flow with weight decay $(1/(2g\beta) \neq 0$ finite) or without weight decay $(1/(2g\beta) = 0)$.

Secondly, the limit $t \to \infty$ and $\beta$ finite yields the stationary distribution of the weights $p(w|\mathcal{D}) \propto \exp\big(-\beta H(w, \mathcal{D})\big)$ (Seung et al., 1992; Naveh et al., 2020), equivalent to the Bayesian posterior for the Gaussian process $f(x) \sim \mathcal{N}(0|\kappa + \beta^{-1}\mathbb{I})$ with kernel

$$\kappa(x, x') = g\,\psi(x)^\top \psi(x') \quad (6)$$

and noisy training samples $y_\mu + \epsilon_\mu$ with Gaussian noise $\epsilon_\mu \overset{\text{i.i.d.}}{\sim} \mathcal{N}(0, \beta^{-1})$ (see Section F for details). And thirdly, considering the dynamics of linear regression is a starting point to study real networks in their lazy learning limits, such as the neural network Gaussian process (Neal, 1995; Williams, 1996), the neural tangent kernel (Jacot et al., 2018), and random feature models (Rahimi & Recht, 2007), which are all linearized forms of real networks.

In addition to the dynamics we are particularly interested in generalization error $\mathcal{L}_{\text{test}}$. It is expected to show little variation across different drawings of train patterns. We will therefore be interested in the average $\langle \mathcal{L}_{\text{test}}\rangle_\mathcal{D}$ over many such drawings to describe the generalization properties of the system. The test error measures the error of the learned predictor $f_*$ made on a new drawing of the data $x_* \sim p, y_* = \bar{w}^\top x_*$, which hence is

$$\langle \mathcal{L}_{\text{test}}\rangle = \frac{1}{2}\big\langle (y_* - f_*)^2\big\rangle_{x_* \sim p(x), y_* = \bar{w} x_*} . \quad (7)$$

## 4. Theory

### 4.1. Background

Following along the lines of (Canatar et al., 2021) we assume that the features of the inputs fulfill the orthogonality relation

$$\delta_{ij}\,\eta_i = \int \psi_i(x)\psi_j(x)\, p(x)\, dx . \quad (8)$$

A common example on a finite set of data points are the principal components of the empirical covariance matrix.

Further we assume a Gaussian distribution on the features $\psi_i \sim \mathcal{N}(0, \eta_i\delta_{ij})$ which are i.i.d. across different training samples $\mu$ for different draws $\psi(x_\mu)$ themselves. While the theory allows for arbitrary $\eta_i$, we assume $\eta_i$ to show a power law decay

$$\eta_i = \eta_1\, i^{-1-\gamma} \quad (9)$$

from here on. Together with the definition of the teacher-student model we can rewrite our training objective (3) as

$$H := \frac{1}{2}\sum_{\mu=1}^{P}\left[\sum_{i=1}^{N}(\bar{w}_i - w_i)\psi_i(x_\mu)\right]^2 + \frac{1}{2g\beta}\sum_{i=1}^{N} w_i^2 \quad (10)$$

$$= \frac{1}{2}\sum_{i,j=1}^{N}(\bar{w}_i - w_i)Q_{ij}(\bar{w}_j - w_j) \quad (11)$$

$$+ \frac{1}{2g\beta}\sum_{i=1}^{N} w_i^2 + \text{const.} , \quad (12)$$

where we dropped terms that are independent of $w$ and defined the feature kernel $Q_{ij} \in \mathbb{R}^{N \times N}$ as

$$Q_{ij} = \sum_{\mu=1}^{P} \psi_i(x_\mu)\psi_j(x_\mu) \quad . \quad (13)$$

Another view on this setting is to consider $\psi$ the eigenmodes of the kernel that describes the Gaussian limit of a neuronal network's output on which the final layer of weights $w$ is trained. The loss (10) implies the gradient dynamics for the coefficients $w_i$

$$\frac{\partial}{\partial t} w_i(t) = \sum_{j=1}^{N} Q_{ij}(\bar{w}_j - w_j) - \frac{1}{g\beta}w_i + \zeta_i(t) , \quad (14)$$

The weights are hence coupled by the matrix $Q$. We will find below that in the limit of high dimensions, we achieve a partial decoupling of the statistics of the modes.

Considering the structure of the right hand side we define $v_i := \bar{w}_i - w_i$ which is the discrepancy between the teacher and the student weight as well as $\Omega_{ij} := Q_{ij} + \delta_{ij}/(g\beta)$ and rewrite the gradient descent as

$$\frac{\partial}{\partial t} v_i(t) = -\sum_{j=1}^{N}\Omega_{ij}v_j(t) + \frac{1}{g\beta}\bar{w}_i + \zeta_i(t) \quad . \quad (15)$$

To investigate the distribution on $v_i(t)$, we utilize the MSRDJ formalism (Martin et al., 1973; De Dominicis, 1976; Janssen, 1976) following the lines of (Helias & Dahmen, 2020) to state a moment-generating functional $Z$ of the

stochastic differential equation

$$Z = \int \mathcal{D}v\,\mathcal{D}\tilde{v}\,\exp\left(S(v,\tilde{v})\right) \tag{16}$$

$$S(v,\tilde{v}) = \int dt \sum_{i=1}^{N} \tilde{v}_i(t)\left(\partial_t v_i(t) + \sum_{j=1}^{N} \Omega_{ij} v_j(t) - \frac{\overline{w}_i}{g\beta}\right) \tag{17}$$

$$+ \beta^{-1} \sum_{i=1}^{N} \tilde{v}_i(t)^2 \,.$$

Ultimately, we are interested in the typical behavior of the system which is invariant under different redraws of the training data set. Our analysis shows that the training process is indeed self-averaging, which means that the dynamics of macroscopic observables such as the test loss, in the limit of sufficiently large systems and data samples, becomes sharply peaked on its typical behavior. Technically, this allows one to consider the average of the moment-generating functional over different draws of the data, which enters in terms of the features in the matrix $Q$ and hence implies the averaged moment generating function

$$\langle Z \rangle_Q = \int \mathcal{D}v\,\mathcal{D}\tilde{v}\,\langle\,\exp\left(S(v,\tilde{v})\right)\rangle_Q\,.$$

We here use that the normalization of the dynamical generating functional $Z$ is independent of $Q$, which allows us to directly average $Z$ (as opposed to $\ln Z$ in the static case) (De Dominicis, 1978; Helias & Dahmen, 2020). By the assumption of Gaussian features $\psi$ in (13), the feature matrix $Q$ is a Wishart matrix. After taking the disorder average, the system can be described with a new action $\bar{S}$ as

$$\langle Z \rangle = \int \mathcal{D}v\mathcal{D}\tilde{v}\,\exp\left(\bar{S}(v,\tilde{v})\right)\quad,$$

whose detailed form is derived in Section A. As the MSRDJ formalism gives us a mapping of an equation of motion to a Gaussian action, the aim is to recover an effective equation of motion from the disorder averaged action $\bar{S}$. To this end, we approximate the distribution $e^{\bar{S}}$ by the Gaussian distribution $\mathcal{N}\left[(\bar{v},\bar{\tilde{v}}),G\right]$ that maximizes the negative Kullback-Leibler (KL) divergence

$$\Gamma(\bar{v},\bar{\tilde{v}},G) := -\mathrm{KL}(\mathcal{N}\left[(\bar{v},\bar{\tilde{v}}),G\right]\|e^{\bar{S}(v,\tilde{v})}) \tag{18}$$

$$= \langle S(v,\tilde{v})\rangle_{(v,\tilde{v})\sim\mathcal{N}\left[(\bar{v},\bar{\tilde{v}}),G\right]} + \frac{1}{2}\ln\det\left[G\right]\quad.$$

The details of this variational Gaussian approximation (VGA) are given in Appendix B and Appendix C. We find that the stationary points of the KL divergence with regard to the parameters $\bar{v}$, $\bar{\tilde{v}}$, and $G$ can be expressed through a

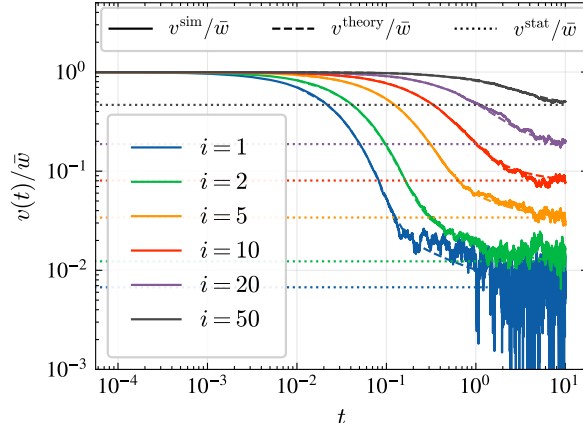

*Figure 1.* Time evolution of normalized mean discrepancy $v_i(t)/\overline{w}_i = (\overline{w}_i - w_i(t))/\overline{w}_i$. Simulation (full curves) compared to theory (dashed curves, (27)). Different curves show different modes $\eta_i$ from blue (large $\eta_i$) to black (small $\eta_i$, see legend).

set of order parameters

$$R(t,s) = \sum_k \eta_k v_k(t)\tilde{v}_k(s)\,,$$

$$\tilde{C}(t,s) = \sum_k \eta_k \tilde{v}_k(t)\tilde{v}_k(s)\,, \tag{19}$$

$$C(t,s) = \sum_k \eta_k v_k(t)v_k(s)\,,$$

that concentrate around their expectation values as $N \to \infty$.

### 4.2. Equations of motion for the first and second order statistics

Computing the stationary point of the KL divergence, we obtain a set of $N$ effective equations of motion that are decoupled across features

$$\left[\partial_t + \frac{1}{g\beta}\right] v_i(t) + P\eta_i \int K(t-s)v_i(s)\mathrm{d}s = \frac{\overline{w}_i}{g\beta} + \xi_i(t), \tag{20}$$

with $i = 1,\ldots,N$ and where we assume the initial condition $w_i(0) = 0$, which corresponds to $v_i(0) = \overline{w}_i$. Despite the processes being statistically independent across features $i$, they interact indirectly because they are driven by uncorrelated centered Gaussian noises $\xi_i$, which, however, share a common and self-consistently determined covariance function

$$\langle \xi_i(t)\xi_j(s)\rangle = \tag{21}$$

$$\delta_{ij}\left[2\beta^{-1}\delta(t-s) + P\eta_i\left\{K * \bar{C} * K^\top\right\}(t,s)\right]. \tag{22}$$

The noise is no longer white, but correlated in time with the correlations mediated through the mean correlation

$\bar{C} := \langle C \rangle$ controlled by the order parameter (19). The temporal memory kernel $K(t,s)$, in addition, mediates a self-coupling of the process with its own past and obeys a linear convolution equation

$$K(t-s) = \int dt' \, \bar{R}(t-t') \, K(t'-s) + \delta(t-s) \quad (23)$$

with $\bar{R} := \langle R \rangle$ and $R$ defined in (19), which can be written as the weighted sum

$$\bar{R}(t-s) = \sum_i \eta_i \, G_i(t-s) \quad (24)$$

where $G_i(t-s) := \langle v(t)\tilde{v}(s) \rangle$ is the Green's function of (20) and obeys itself

$$-\left[\partial_t + \frac{1}{g\beta}\right] G_i(t-s) \quad (25)$$

$$= P\eta_i \int du \, K(t-u) \, G_i(u-s) + \delta(t-s) \,.$$

By their linearity and by the appearing convolutions, this set of equations is conveniently solved in the Fourier domain $t-s \to \omega$ as $\hat{K}(\omega) = \left[1 - \hat{R}(\omega)\right]^{-1}$, $\hat{R}(\omega) = \sum_i \eta_i \, \hat{G}_i(\omega)$, and

$$\hat{G}_i(\omega) = -\left[i\omega + \frac{1}{g\beta} + P\eta_i \, \hat{K}(\omega)\right]^{-1}. \quad (26)$$

The equation of motion for the mean is obtained by taking the expectation value over the noise of (20), which yields

$$\left[\partial_t + \frac{1}{g\beta}\right] \langle v_i(t) \rangle + P\eta_i \int K(t-s) \langle v_i(s) \rangle \, ds = \frac{\bar{w}_i}{g\beta}, \quad (27)$$

where the initial condition is $\langle v_i(0) \rangle = \bar{w}_i$. This expression shows that the time-scale of the $i$-th mode is controlled by the self-interaction term $P\eta_i K * \langle v \rangle$, which hence depends inversely on the $i$-th eigenvalue $\eta_i$. This timescale controls how fast the corresponding mode is learned. Large eigenmodes therefore approach the teacher signal more quickly than slow ones as can be seen in Fig. 1 where the time evolution of the mean discrepancies for different modes is displayed in relation to the theoretical mean value $\langle v \rangle$ given by (27). Likewise, the amount of data enters here multiplicatively as $P$, showing that more data leads to faster learning of the teacher, as intuitively expected. The same is true for the stationary value of $\langle v \rangle$, which is also determined through the self-interaction term as

$$\lim_{t \to \infty} \langle v \rangle_i(t) = \frac{\bar{w}}{1 + g\beta P\eta_i \int_0^\infty K(t) \, dt} \,. \quad (28)$$

Its final value is thus influenced through the kernel eigenvalue $\eta_i$, that data size $P$, as well as regularization strength $g\beta$ in the same manner. Given the fact that $K$ acts as a mean

field coupling, it however depends by (23) and (24) on the entire distribution of eigenvalues instead of a single one.

In addition, it can be observed that the fluctuations of $v$ around the mean $\langle v(t) \rangle$ increase over time. Their covariance follows from (20) as the Green's function (25) convolved with the noise $\xi$, which hence yields

$$\langle \delta v_i(t) \delta v_i(s) \rangle$$

$$= 2/\beta \int_0^{\min(t,s)} du \, G_i(t-u) G_i(s-u) \quad (29)$$

$$+ P\eta_i \left[G_i * K * \bar{C} * K^\top * G_i^\top\right](t,s), \quad (30)$$

where

$$\left[K * \bar{C} * K^\top\right](u,u')$$

$$= \int_0^u dt \int_0^{u'} ds \, K(u-t) \, \bar{C}(t,s) \, K(u'-s),$$

which are solved self-consistently with $\bar{C}$, which, in turn, requires the covariance $\langle \delta v_i(t) \delta v_i(s) \rangle$ and the mean given by (61). The fluctuations are both dependent on the initial Langevin noise $\zeta$ and on the correlation, which leads to a self-reinforcement over time whose consequences are discussed in the following chapter.

### 4.3. Time dynamics of the generalization error

The generalization error (7) can be determined very naturally in our theory as it can be expressed in terms of one of the order parameters as

$$\mathcal{L}_{\text{test}} = \frac{1}{2} \left\langle \sum_{i=1}^N \psi_i \left\langle (\bar{w}_i - w_i)^2 \right\rangle_\zeta \right\rangle_{\psi \sim \mathcal{N}(0, \eta_i \delta_{ij})} \quad (31)$$

$$= \frac{1}{2} \sum_{i=1}^N \eta_i \langle v_i(t)^2 \rangle_\zeta = \frac{1}{2} \bar{C}(t,t) \quad,$$

allowing us to compute it together with (30). Fig. 2(a) shows the time evolution of the test error as a function of time for different temperatures $\beta^{-1}$. As the noise variance $2\beta^{-1}$ increases, the generalization worsens as expected. For small times, the error declines, but eventually the error increases again after reaching a local minimum. To explain this behavior we decompose the test error

$$\mathcal{L}_{\text{test}} = \frac{1}{2} \sum_{i=1}^N \eta_i \left[\underbrace{\langle v_i(t) \rangle^2}_{\text{bias}} + \underbrace{\langle \delta v^2(t) \rangle}_{\text{variance}}\right], \quad (32)$$

into the bias and the variance contribution. As seen in Fig. 2(b), the bias term decays in a monotonic fashion, independent of $\beta$, while the fluctuations grow with higher temperatures and, so does the variance term. In Fig. 2(b) we see the minimum formed by this temporal bias-variance

(a)
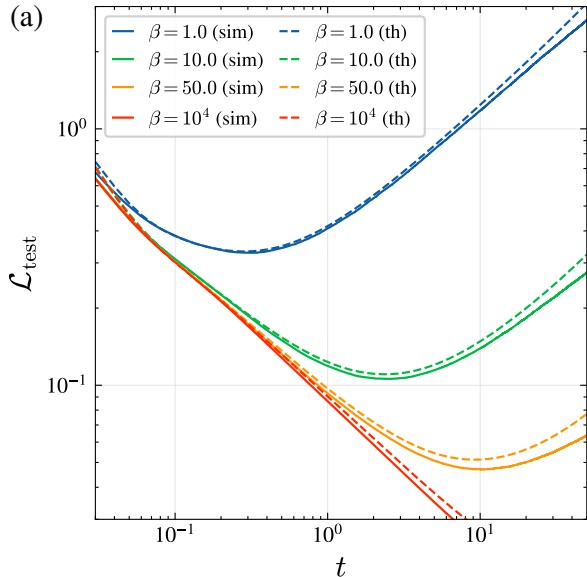

(b)
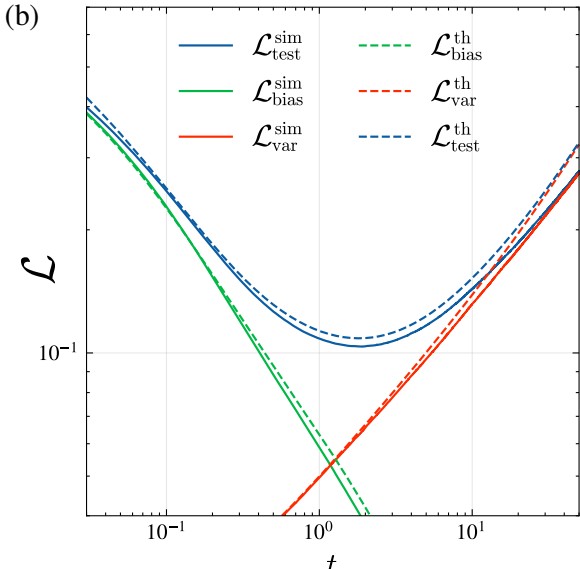

*Figure 2.* (a) Test error for different strengths $2\beta^{-1}$ of the dynamic noise with $\beta = 10000$ (red), $\beta = 50$ (green) and $\beta = 10$ (blue). The solid curves show the theory (31), the dashed lines the simulation. The other parameters are $g\beta = 10^3, P = N = 100, \Lambda_{ij} = i^{-3/2}\delta_{ij}$. The time step used for the simulation is $\mathrm{d}t = 10^{-4}$, for the theory it is $\mathrm{d}t = 10^{-2}$. The disorder average in the simulation is taken over $10^5$ different realizations of training data. (b) Bias-variance decomposition of the test error. The blue curves show the full test error (31) (full curve simulation; dashed curve theory), the green curves show the bias contribution to the test loss, the red show the variance part (cf. (32)). The expectation value of the kernel follows the power law $\Lambda_{ij} = i^{-3/2}\delta_{ij}$. The time step used for the simulation is $\mathrm{d}t = 10^{-4}$, for the theory it is $\mathrm{d}t = 10^{-2}$. Disorder average in simulation taken over $10^5$ different realizations of training data sets. Other parameters $\beta = 10$ and $g\beta = 10^3$, system at the interpolation threshold ($P = N = 10^2$).

trade-off. While initially the error is dominated by the decrease of the large modes' discrepancies between student and teacher weights, the variance adds up over time, eventually becoming the dominant contribution. This result explains the practical use of early stopping, i.e. aborting the training before equilibrium is reached, to be beneficial in the presence of significant noise. As this observation is obtained with weak regularization, it raises the question of whether this behavior is altered, if the weight decay is regularized more strongly.

The L2-regularization term $1/(g\beta)$ in the loss function (3) favors small weights and ensures the convergence of the weights even for singular feature-feature matrices (13). It also prevents the weights from accumulating noise, acting as a cutoff for the variance part of the generalization error. Fig. 3 shows the test error for high but fixed noise and varying regularization. Larger regularization $1/(g\beta)$ prevents the generalization error from diverging. Interestingly, while this affects larger times, it has small effect on the location and the magnitude of the minimum loss. In a heuristic fashion this can again be explained with the bias-variance trade-off: As seen in Fig. 1 the decay of the bias term is mainly governed by the fast modes, corresponding to the large eigenvalues that adapt the strongest to data. As the regularization counteracts the $P\eta K*$-term in the equation

of motion (20) it affects those weights stronger that have a small eigenvalue $\eta$; on short time-scales the bias term is only weakly affected. The variance, on the other hand, is a collective phenomenon. While the fast modes again contribute on shorter timescales, because of the spectral bias, the slow modes become more important over time because of their long range Green's functions in (30) which counteract the small eigenvalues. Their decay is, however, enforced by a larger regularization as can be seen in (26) and therefore has a restraining effect on the variance part. This, in turn, limits the generalization error on larger timescales. While the generalization is expected to get worse if the regularization is too high, this result suggests that regularization can be particularly helpful for training in the presence of noise.

In the noiseless case our setup is identical to the one (Canatar et al., 2021) consider in the stationary case and it can be shown that in the limit $t \to \infty$, (31) recovers their results (see Section I).

## 5. Neural Scaling Laws

We utilize the previously derived results to analyze the thermodynamic limit of the model, which is known to exhibit power-laws. As the full generalization error in general is a superposition of multiple power-laws, it is beneficial to look

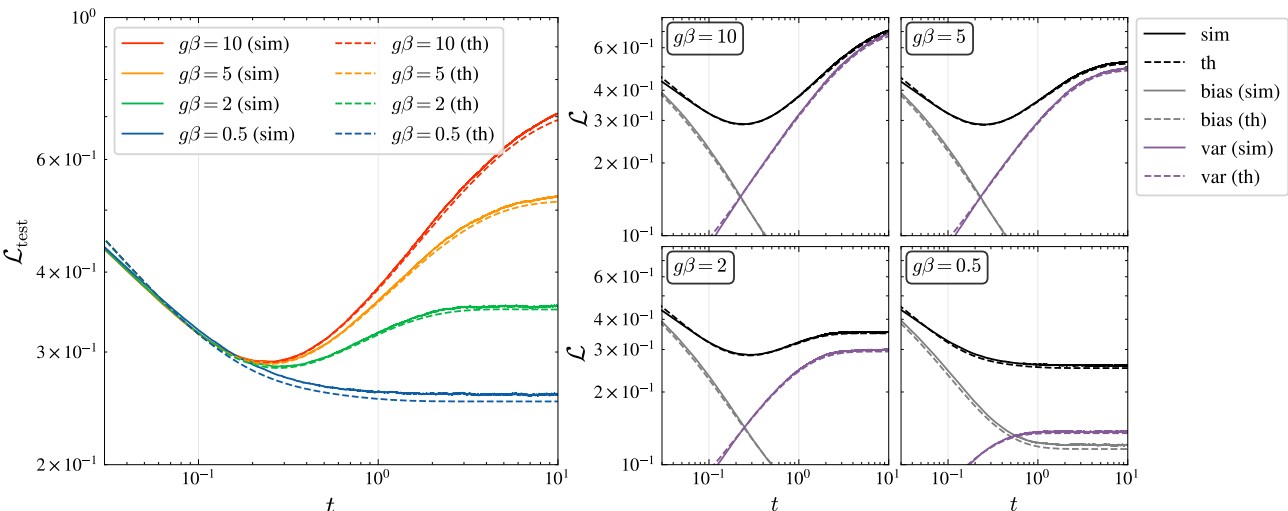

*Figure 3.* Effect of regularization on early stopping. Left panel: Test error for different strengths $g\beta$ of the regularization. The dashed curves show the theory, the solid curves the simulation. Right panel: Individual test error curves with their respective bias-variance decomposition. Dashed curves are theory, solid curves simulation. Other parameters for all panels $\beta = 10^0$, $P = N = 100$, $\Lambda_{ij} = i^{-3/2}\delta_{ij}$. The time step used for the simulation was $dt = 10^{-4}$, for the theory it is $dt = 10^{-2}$. Disorder average in simulation taken over $10^5$ different realizations of training data sets.

at each component individually. A detailed derivation of the power-laws given below can be found in Section G.

### 5.1. Bias error

The bias error is bounded by the limiting Ornstein-Uhlenbeck process, which is obtained from (25) by approximating $K(t-s) \simeq \delta(t-s)$. This approximation corresponds to the neglect of fluctuations of $Q$, replacing it by $\langle Q \rangle$ in (14). Since fluctuation corrections slow down the learning dynamics, the bias will thus decay slower than

$$\mathcal{L}_{\text{bias,OUP}}(t) = \frac{1}{2} \sum \eta_i \bar{w}_i^2 e^{-2\left(P\eta_i + \frac{1}{g\beta}\right)t} \quad (33)$$

which in the $P, N \gg 1$ limit decays as (see Section G)

$$\mathcal{L}_{\text{bias,OUP}} \sim c_1 (Pt)^{-\alpha/(1+\alpha)} - c_2 N^{-\alpha} \quad (34)$$

for $c_1$ and $c_2$ being positive constants. For constant $N$ this corresponds to the well known pareto frontier as function of compute $f = 6PNt$ as shown in Fig. 4. For $P, t \to \infty$, this recovers the result from (Bordelon et al., 2024) that $\mathcal{L} \sim N^{-\alpha}$.

### 5.2. Variance error

To investigate the behavior of the system in the presence of large noise it is natural to look at the variance error in the limit $\beta \to 0$. In this regime, $\mathcal{L}_{\text{var}}$ can be approximated as

$$\mathcal{L}_{\text{var}}(t) \approx \mathcal{L}_\beta(t)$$

$$:= \frac{1}{\beta} \sum_{i=1}^{N} \eta_i \int_0^t ds\, G_i(t-s)^2 \quad .$$

Taking into account the same considerations as for $\mathcal{L}_{\text{bias}}$ we find that

$$\mathcal{L}_\beta \sim c_3 \beta^{-1} P^{-\alpha/(1+\alpha)} t^{1/(1+\alpha)} \quad , \quad (35)$$

with $c_3 = (1+\alpha)c_1$. This is in accordance with our intuition that the variance decreases with the amount of training data and that the noise accumulates and hence grows over time. Interestingly, we find the same power-law as a function of $P$ as for the bias.

Fig. 4 shows the time evolution of $\mathcal{L}$ as a function of $Pt$ for different values of $P$. While the theoretical predicted power-laws agree nicely, the $N^{-\alpha}$ correction cannot be neglected and for finite system sizes the bias part has to be viewed as a superposition of power-laws.

### 5.3. Early stopping

The optimal stopping time $t^*$ is given by the minimum of the test error through

$$0 = \left. \frac{d\left(\mathcal{L}_{\text{bias}}(t) + \mathcal{L}_{\text{var}}(t)\right)}{dt} \right|_{t=t^*} \quad .$$

Evaluating this equation, we find

$$t^* = \frac{\beta}{2} \frac{\alpha}{1+\alpha} \quad (36)$$

as well as for the generalization error evaluated at that point

$$\mathcal{L}_{\text{test}}(t^*) \sim d_1 (P\beta)^{-\alpha/(1+\alpha)} - d_2 N^{-\alpha} \quad . \quad (37)$$

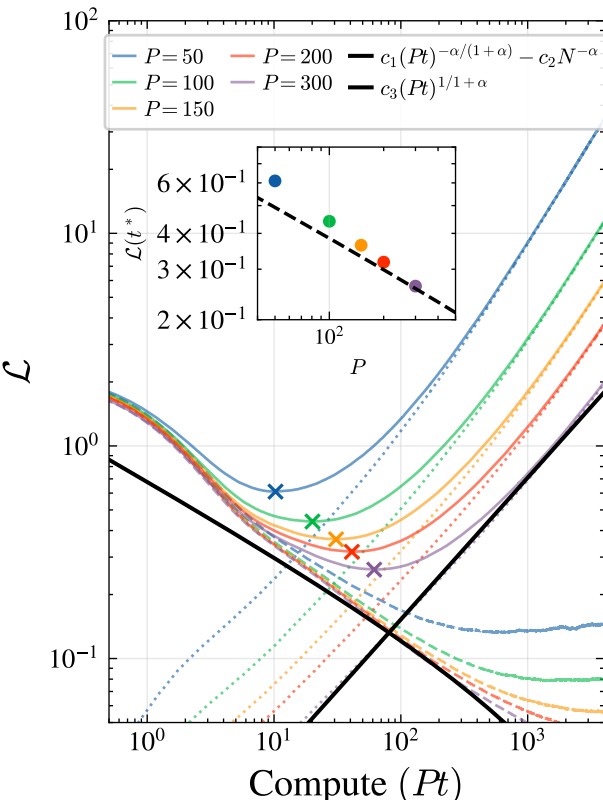

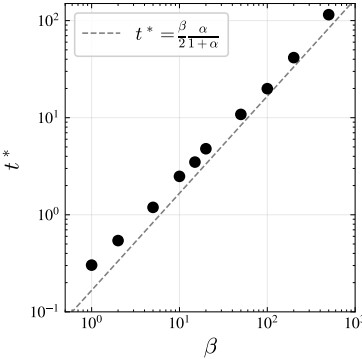

*Figure 5.* Optimal stopping time as function of $\beta$. The points represent the simulation, the dashed gray line the theoretical estimate based on (36). Other parameters: $P = N = 100$, $\Lambda_{ij} = i^{-3/2}\delta_{ij}$, $g\beta = 10^3$. The time step used for the simulation was $\mathrm{d}t = 10^{-4}$. Disorder average in simulation taken over $5 \times 10^3$ different realizations of training data sets.

## 6. Discussion

In this work we present a self-consistent dynamical mean-field theory to describe the training dynamics of kernel regression. The framework is able to cover both gradient flow as well as Langevin stochastic gradient descent and, at equilibrium, makes the link to Bayesian inference of learning. Here we obtain an expression for the dynamics of the generalization error that is applicable to the relevant case of power-law distributed kernel spectra, as they are frequently encountered in real data and in the kernel regime of trained networks (Lee et al., 2018; Jacot et al., 2018). We demonstrate that this setting shows the phenomenon of early stopping, where the test-loss is minimized before training has converged. This work hence effectively extends on prior research such as (Krogh & Hertz, 1992) and (Advani et al., 2020), who studied the case of learning dynamics on Gaussian i.i.d. data and (Canatar et al., 2021), who studied the case of kernel regression after convergence and whose results can be obtained from our theory.

The theory shows that the problem of learning is self-averaging, implying that the data-averaged behavior is close to the behavior of a system trained on a single data set. This allows us to average the dynamical partition function over data disorder and the noise of Langevin training. A technical advantage of the dynamical approach is that it avoids the use of replicas (De Dominicis, 1978). We find that in the limit of large dimension $N \to \infty$, the full statistics are entirely described by two self-averaging order parameters, the trace of the system's response function and the trace of the auto-correlation of the parameters. The latter, moreover, is proportional to the test error. This simplicity is due to averaging over the kernel directly, while previous work relied on additional auxiliary fields (Bordelon et al., 2024). Our work thus yields a compact and interpretable description of

*Figure 4.* Neural scaling law for the test error as a function of training time arises from superposition of exponentially decaying modes (33). Solid colored curves are the test error while the colored dashed and dotted curves display the bias and variance error, respectively, for different sample sizes $P$ obtained from numerical experiments. The black descending curve is the limiting case for the bias part (34), the ascending black curve the variance approximation (35). The other parameters are $\beta = 10^0$, $g\beta = 10^5$, $N = 1000$, $\Lambda_{ij} = i^{-3/2}\delta_{ij}$, $\mathrm{d}t = 0.005$. Disorder average in simulation taken over $10^2$ different realizations of training data sets. The inset plot shows the minimal loss as a function of $P$. The colored dots correspond to the crosses in the main plot, the dashed line is the power-law approximation given in (37).

We observe that the optimal stopping time $t^*$ is independent of both $P$ and $N$ but is only a function of $\beta$, a result that can be verified in numerical simulations (see Fig. 5). This simple linear relationship (36) should be considered in conjunction with (37), and may be interpreted as follows: under high-noise conditions, learning must be terminated early to prevent the accumulation of noise, since further training cannot recover additional structure from the data. We further see in (37) that the amount of training data acts in a conjugate way to the noise $\beta$. We see in Fig. 4 that this relationship holds approximately in simulations, however finite size effects play an important role in correctly estimating the optimal generalization error $\mathcal{L}_{\text{test}}(t^*)$.

the dynamics while still maintaining the full generality of the problem. In the large $N$ limit, the central limit theorem guarantees the Gaussianity of the process, which renders the performed variational Gaussian approximation exact (see also Appendix C). The fact, that the test error appears in a natural way in the theory makes it a particularly useful description to investigate the generalization performance.

From the Gaussian action we derive an effective stochastic equation of motion that exposes the spectral dependence of the mean evolution of each mode on its eigenvalue. It yields a dynamical view on the spectral bias of learning (Canatar et al., 2021), showing that larger eigenmodes are not only learned with higher accuracy, but are also learned earlier than smaller ones. In addition, we show that the effective equations of motion of all modes become mutually statistically independent. The relaxation time of each mode is slowed down by a collective temporal non-local coupling that is caused by the disorder average and whose temporal shape depends on the dynamics of all remaining modes. This coupling can be identified as a geometric series of the system's perturbation response in frequency domain and acts like an Onsager reaction term for the system (Fischer & Hertz, 1991). Even though modes become statistically independent, they are coupled indirectly through this memory kernel: thus, modes that are learned more slowly, indirectly also slow down the learning process of all remaining modes.

The joint dataset formally acts as a collective disorder which in addition produces an effective noise term in the equation of motion. This term, too, mediates an indirect coupling among modes, as the mean-discrepancy of each mode contributes additively to this noise. This temporally correlated and time-dependent noise, in turn, increases the variance of the individual weights. As the effective noise is linked to the generalization error, we find the full dynamics of the bias-variance decomposition of the test error. It shows that the optimal duration for learning that yields the minimal test error (early stopping) results from an interplay between the effective noise, controlling the gradual growth of the variance term of the loss, and the decay time of the response function, which governs the decline of the bias term of the generalization error. This early stopping time, in the thermodynamic limit, becomes a constant that only depends on the decay exponent of the kernel's eigenmodes and the inverse temperature. It can be used to derive an expression for the optimal generalization error as a superposition of competing scaling laws stemming from the bias and variance terms respectively.

Our setup can easily be extended to study the influence of other sources of noise an example being static label noise as discussed in Section H. While it is restricted to linearized settings, such as kernel ridge regression or linear regression, further work is needed to include a combination of the dynamical formalism we developed here with contemporary work on feature learning in non-linear networks (Li & Sompolinsky, 2021; Seroussi & Ringel, 2023; Pacelli et al., 2023; Rubin et al., 2025; Lauditi et al., 2025) to describe the learning dynamics of networks closer to real-life settings and a possible extension is discussed for the case of committee machines in Section J. Recent work, however, shows that the kernel limit of networks may indeed capture the universal behavior in the large data limit (Coppola et al., 2026).

## Impact Statement

This paper presents work whose goal is to advance the field of machine learning. The work in particular aims to develop the theoretical understanding of learning on high-dimensional data, with ubiquitous properties. A fundamental understanding of these properties is helpful to provide guarantees for learning and to guide future improvements of algorithms and architectures.

## Acknowledgements

This work was partly funded by the Deutsche Forschungsgemeinschaft (DFG, German Research Foundation) - 368482240/GRK2416, the Helmholtz Association Initiative and Networking Fund under project number SO-092 (Advanced Computing Architectures, ACA), the Deutsche Forschungsgemeinschaft (DFG, German Research Foundation) as part of the SPP 2205 – 533396241, and the DFG grant 561027837/HE 9032/4-1. Open access publication funded by the Deutsche Forschungsgemeinschaft (DFG, German Research Foundation) – 491111487. The authors gratefully acknowledge the computing time granted by the JARA Vergabegremium and provided on the JARA Partition part of the supercomputer JURECA at Forschungszentrum Jülich (computation grant JINB33).

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

# Appendix

## A. Generating functional in MSRDJ formalism

Starting from the equation of motion in 15 we want to investigate the distribution on $v_i(t)$, to which purpose we utilize the MSRDJ formalism (Helias & Dahmen, 2020) to state a functional $Z$ which generates the functional moments of the stochastic differential equation

$$Z = \int \mathcal{D}v\mathcal{D}\tilde{v} \, \exp\left(S(v,\tilde{v})\right) \tag{38}$$

$$S(v,\tilde{v}) = \int dt \sum_{i=1}^{N} \tilde{v}_i(t)\left(\partial_t v_i(t) + \sum_{j=1}^{N} \Omega_{ij} v_j(t) - \frac{\overline{w}_i}{g\beta}\right) + \frac{1}{\beta}\sum_{i=1}^{N} \tilde{v}_i(t)^2. \tag{39}$$

with $\Omega_{ij} := Q_{ij} + \delta_{ij}/(g\beta)$. As we are ultimately interested in the disorder averaged learning statistics on $v$ we rather need to consider $\langle Z \rangle$ and its derivatives. Considering the fact that, due to the Gaussianity of the features $\psi$, the feature matrix $Q$ is a Wishart matrix, we can write the disorder-averaged moment generating functional as

$$\langle Z \rangle_Q = \int \mathcal{D}v\mathcal{D}\tilde{v} \, \langle \exp(S(v,\tilde{v})) \rangle_Q \tag{40}$$

$$= \int \mathcal{D}v\mathcal{D}\tilde{v} \, \exp\left(\bar{S}(v,\tilde{v})\right) \tag{41}$$

with

$$\bar{S}(v,\tilde{v}) = \int dt \sum \tilde{v}_i(t)\left(\left[\partial_t + \frac{1}{g\beta}\right]v_i(t) - \frac{\overline{w}_i}{g\beta} + \frac{1}{\beta}\sum_{i=1}^{N}\tilde{v}_i(t)^2\right) - \frac{P}{2}\ln\det\left(\mathbb{I} - \Lambda\left[\int dt\, \tilde{v}(t)v(t)^\top + v(t)\tilde{v}(t)^\top\right]\right) \tag{42}$$

with $\mathbb{I}_{ij} = \delta_{ij}$. The last line is the cumulant-generating function of the Wishart distribution; it can be obtained by a trivial Gaussian integral over the Gaussian distributed modes $\psi$. We here introduced the diagonal matrix $\Lambda_{ij} = \delta_{ij}\eta_i$.

This action is analytically intractable. However, we will be able to treat this term analytically in the large $N$ limit by first introducing a number of order parameters and subsequently performing a variational Gaussian approximation (VGA), which will approximate the process $v$ by a Gaussian with non-zero mean. We will present this approach in the following two sections.

## B. Identification of order parameters

The term after disorder in (42), rewritten as $-\frac{P}{2}\text{tr}\ln\left(\mathbb{I} - \Lambda\left[\int dt\, \tilde{v}(t)v(t)^\top + v(t)\tilde{v}(t)^\top\right]\right)$ and expanded into the power series $\ln(1-x) = -\sum_{n=1}^{\infty}\frac{x^n}{n}$ yields with the short hand $J = \int \tilde{v}(t)v(t)^\top dt$

$$-\frac{1}{2}\ln\det\left(\mathbb{I} - \Lambda\left(J + J^\top\right)\right) \tag{43}$$

$$= \frac{1}{2}\sum_{n=1}^{\infty}\frac{1}{n}\left[\Lambda\left(J + J^\top\right)\right]^n \tag{44}$$

$$= \frac{1}{2}\sum_i \eta_i J_{ii} + \sum_i \eta_i J_{ii}^\top$$

$$+ \frac{1}{4}\sum_i\sum_j \eta_i J_{ij}\,\eta_j J_{ji} + \frac{1}{4}\sum_i\sum_j \eta_i J_{ij}\,\eta_j J_{ji}^\top + \frac{1}{4}\sum_i\sum_j \eta_i J_{ij}^\top\,\eta_j J_{ji} + \frac{1}{4}\sum_i\sum_i \eta_i J_{ij}^\top\,\eta_j J_{ji}^\top$$

$$+ \dots.$$

In this product the order of $J$ and $J^\top$ determines which self-averaging terms arise, namely one has

$$\left[\Lambda J\,\Lambda J\right]_{ij} \equiv \iint \eta_i\tilde{v}_i(t) \underbrace{\sum_k \eta_k v_k(t)\,\tilde{v}_k(s)}_{=:R(t,s)}\, v_j(s)\, dt\, ds\,, \tag{45}$$

$$\left[\Lambda J^\top \Lambda J^\top\right]_{ij} \equiv \iint \eta_i v_i(t) \underbrace{\sum_k \eta_k\tilde{v}_k(t)\, v_k(s)}_{=R(s,t)}\, \tilde{v}_j(s)\, dt\, ds\,,$$

$$\left[\Lambda J^\top \Lambda J\right]_{ij} \equiv \iint \eta_i v_i(t) \underbrace{\sum_k \eta_k\tilde{v}_k(t)\,\tilde{v}_k(s)}_{=:\tilde{C}(t,s)}\, v_j(s)\, dt\, ds\,,$$

$$\left[\Lambda J\,\Lambda J^\top\right]_{ij} \equiv \iint \eta_i\tilde{v}_i(t) \underbrace{\sum_k \eta_k v_k(t)\, v_k(s)}_{=:C(t,s)}\, \tilde{v}_i(s)\, dt\, ds\,.$$

So we obtain the rules

$$\Lambda J \Lambda J \to \eta\, \tilde{v}\, R\, v\,,$$
$$\Lambda J^\top \Lambda J^\top \to \eta\, v\, R^\top\, \tilde{v}\,,$$
$$\Lambda J^\top \Lambda J \to \eta\, v\, \tilde{C}\, v\,,$$
$$\Lambda J \Lambda J^\top \to \eta\, \tilde{v}\, C\, \tilde{v}.$$

One notices that the remaining terms in (45), if written under the trace and after replacing the indicated parts by the self-averaging quantity, mediate a coupling only between modes of the same index, so they lead to an action that is diagonal.

In the limit $N \to \infty$, we will find that $C$ and $R$ concentrate to their mean value. To demonstrate this, one needs to show that the statistics of $(\tilde{v}_k, v_k)$ becomes independent across $k$, as we will show in the following sections. If the distribution of the $(\tilde{v}_k, v_k)$ factorizes across $k$, the concentration of $C$ and $R$ follows from the central limit theorem. One additional point to note here is that the power law $\eta_k = k^{-(1+\alpha)}$ is such that

$$\sum_{k=1}^{N} \eta_k \simeq \int_1^N k^{-(1+\alpha)}\, dk = -\alpha^{-1}\, k^{-\alpha}\Big|_1^N \overset{N\to\infty}{\Rightarrow} \alpha^{-1} \tag{46}$$

converges to a constant for the assumed exponents $\alpha > 0$. We will find below that the correlation functions $\langle v_k v_k\rangle$ and response functions $\langle \tilde{v}_k v_k\rangle$ also decline with $k$, which is due to the appearance of $\eta_k$ in the effective equations of motion (69), so that (46) is a majorization of the definitions of $C$ and $R$ and thus both also exist in the limit $N \to \infty$.

## C. Variational Gaussian approximation

To treat the data-dependent term, one may perform a variational Gaussian approximation (VGA) were modes are pairwise independent, but may have mode-dependent variance and response functions jointly contained in $G$ as well as a mode-dependent mean $(\bar{v}, \bar{\tilde{v}})$.

This motivates the variational ansatz

$$(v, \tilde{v}) \sim \mathcal{N}\left[(\bar{v}, \bar{\tilde{v}}), G\right] \propto \exp\left\{-\frac{1}{2}\begin{pmatrix} v - \bar{v} \\ \tilde{v} - \bar{\tilde{v}} \end{pmatrix}^\top G^{-1} \begin{pmatrix} v - \bar{v} \\ \tilde{v} - \bar{\tilde{v}} \end{pmatrix}\right\} \tag{47}$$

$$G = \begin{pmatrix} G^{(vv)} & G^{(v\tilde{v})} \\ G^{(\tilde{v}v)} & G^{(\tilde{v}\tilde{v})} \end{pmatrix}\,, \tag{48}$$

where each $G^{(xy)}$ is a diagonal $N \times N$ matrix, with elements $G_{ij}^{(xy)}(t,s) = \delta_{ij} G_i^{(xy)}(t,s)$ for $x, y \in \{v, \tilde{v}\}$. We will show a posteriori that this ansatz becomes exact in the limit $N \to \infty$.

The parameters $\bar{v}$, $\bar{\tilde{v}}$, and $G$ will be determined from the equation of state, which arises from minimizing the KL divergence between $p(v, \tilde{v}) \propto e^{S(v,\tilde{v})}$ and the Gaussian measure (47)

$$\Gamma(\bar{v}, \bar{\tilde{v}}, G) := -\mathrm{KL}\left(\mathcal{N}\left[(\bar{v}, \bar{\tilde{v}}), G\right]\,\Big\|\, e^{\bar{S}(v,\tilde{v})}\right)$$

$$= \left\langle \bar{S}(v, \tilde{v})\right\rangle_{(v,\tilde{v})\sim\mathcal{N}\left[(\bar{v},\bar{\tilde{v}}),G\right]} + \frac{1}{2}\ln\det[G],$$

where the latter term is the entropy of the Gaussian. The parameters $\bar{v}$, $\bar{\tilde{v}}$, and $G$ that minimize the KL divergence are determined by the stationarity conditions, also known as equations of state,

$$0 \overset{!}{=} \frac{\delta}{\delta\{\bar{v}, \bar{\tilde{v}}\}}\left[\left\langle \bar{S}(v, \tilde{v})\right\rangle_{(v,\tilde{v})\sim\mathcal{N}\left[(\bar{v},\bar{\tilde{v}}),G\right]}\right]\Bigg|_{G^{(\tilde{v}\tilde{v})}=0, \bar{\tilde{v}}=0}, \tag{49}$$

$$0 \overset{!}{=} \frac{\delta}{\delta G^{(xy)}}\left[\left\langle \bar{S}(v, \tilde{v})\right\rangle_{(v,\tilde{v})\sim\mathcal{N}\left[(\bar{v},\bar{\tilde{v}}),G\right]} + \frac{1}{2}\ln\det[G]\right]\Bigg|_{G^{(\tilde{v}\tilde{v})}=0, \bar{\tilde{v}}=0}, \tag{50}$$

where we need to assure that $\bar{\tilde{v}} = 0$ as well as $G^{(\tilde{v}\tilde{v})} \equiv 0$.

For the equations of state, we also need the inverse of the matrix (48), which, due to its diagonality in $i$, can be inverted for each $i$ separately as

$$\frac{\delta}{\delta G_i^{(xy)}(t,s)}\frac{1}{2}\ln\det[G]\Big|_{G^{(\tilde{v}\tilde{v})}=0} = \frac{1}{2}[G]_i^{-1}(t,s) \tag{51}$$

$$\text{with}\quad [G]^{-1} = \begin{pmatrix} 0 & \left[G^{(\tilde{v}v)}\right]^{-1} \\ \left[G^{(v\tilde{v})}\right]^{-1} & -\left[G^{(v\tilde{v})}\right]^{-1}G^{(vv)}\left[G^{(\tilde{v}v)}\right]^{-1} \end{pmatrix}.$$

Applied to (45), extended to arbitrary orders of $n$, this means we only need to take into account terms of the form

$$\left\langle \mathrm{tr}(\Lambda J)^n\right\rangle_{(v,\tilde{v})\sim\mathcal{N}\left[(\bar{v},\bar{\tilde{v}}),G\right]} \to [\bar{R}]^n, \tag{52}$$

$$\left\langle \mathrm{tr}(\Lambda J^\top)^n\right\rangle_{(v,\tilde{v})\sim\mathcal{N}\left[(\bar{v},\bar{\tilde{v}}),G\right]} \to [\bar{R}^\top]^n,$$

$$\left\langle \mathrm{tr}(\Lambda J)^{n-k}(\Lambda J^\top)^k\right\rangle_{(v,\tilde{v})\sim\mathcal{N}\left[(\bar{v},\bar{\tilde{v}}),G\right]} \overset{1\le k\le n-1}{\Rightarrow} [\bar{R}]^{n-k-1}\bar{C}[\bar{R}^\top]^{k-1}\bar{\bar{C}},$$

where

$$\bar{R}^\top(s,t) = \bar{R}(t,s) = \sum_i \eta_i\left[\bar{v}_i(t)\bar{\tilde{v}}_i(s) + G_i^{(v\tilde{v})}(t,s)\right], \tag{53}$$

$$\bar{C}(t,s) = \sum_i \eta_i\left[\bar{v}_i(t)\bar{v}_i(s) + G_i^{(vv)}(t,s)\right],$$

$$\bar{\bar{C}}(t,s) = \sum_i \eta_i\left[\bar{\tilde{v}}_i(t)\bar{\tilde{v}}_i(s) + G_i^{(\tilde{v}\tilde{v})}(t,s)\right].$$

We may study explicitly the terms that we have dropped, when writing the replacement rules (52) to show that they either vanish completely or are subleading in the $N \to \infty$ limit. Consider the example of $n = 2$

$$\mathrm{tr}\left[\Lambda J \Lambda J\right] \equiv \sum_{ik} \eta_i\eta_k \iint \tilde{v}_i(t)v_k(t)\,\tilde{v}_k(s)v_i(s)\,\mathrm{d}t\,\mathrm{d}s. \tag{54}$$

A priori, there are two qualitatively different contributions. It is easiest to first consider them perturbatively and subsequently perform an infinite resummation of all such contributions, which we technically do with the VGA. The first perturbative correction is of the form

$$\sum_{ik} \eta_i \eta_k \int \tilde{v}_i(t) v_k(t) \, \mathrm{d}t \int \langle \tilde{v}_k(s) v_i(s) \rangle \, \mathrm{d}s \,,$$

which in principle presents a coupling term between modes $(i, k)$, mediated by the time-averaged response function $\int \langle \tilde{v}_k(s) v_i(s) \rangle \, \mathrm{d}s$. The latter response function, however, vanishes, because in the Ito-convention employed here, equal time response functions vanish (see, e.g., (Coolen, 2001; Helias & Dahmen, 2020)). The second perturbative contribution is

$$\sum_i \eta_i \iint \tilde{v}_i(t) \sum_k \eta_k \langle v_k(t) \, \tilde{v}_k(s) \rangle v_i(s) \, \mathrm{d}t \, \mathrm{d}s \,,$$

which is non-zero, because for $t > s$ the response function $\langle v_k(t) \, \tilde{v}_k(s) \rangle \neq 0$ in general. Importantly, this term only mediates a self-coupling of mode $i$ $(v_i)$ to itself $(\tilde{v}_i)$, thus it will not cause correlations between modes. This term is the one taken into account by the first replacement rule (52).

Analogous considerations hold for all higher powers $n > 2$, where we get products of fields $\tilde{v}_i(r) v_j(r) \, \tilde{v}_j(s) v_k(s) \ldots \tilde{v}_l(t) v_i(t)$: Whenever two fields with different mode indices are paired, e.g. $\langle \tilde{v}_i(r) v_k(s) \rangle$ this implies $r < s$ for the contribution not to vanish. But contracting all remaining fields in pairs one always also gets a pairing such as $\langle v_j(r) \tilde{v}_j(s) \rangle$, which requires $r > s$, so together it will lead to vanishing contribution.

An additional perturbative contribution comes from pairing two fields $v$. As a contribution to (54) one has

$$\sum_{ik} \eta_i \eta_k \iint \tilde{v}_i(t) \, \tilde{v}_k(s) \, \langle v_k(t) v_i(s) \rangle \, \mathrm{d}t \, \mathrm{d}s \,, \tag{55}$$

which, for any pair of modes $(i, k)$ contributes a correlated Gaussian noise (due to the appearance of $\tilde{v}_i(t) \, \tilde{v}_k(s)$) in proportion to the correlation $\langle v_k(t) v_i(s) \rangle$ itself. Note first, that a consistent solution is therefore one where modes are uncorrelated. Note second that the driving noise $\zeta_i$ is uncorrelated across modes, so it does not drive these correlations, so that vanishing correlations are indeed the state assumed. Moreover, a single term in the sum $\sum_k$ contributes this correlation. We may compare its magnitude to the magnitude that arises from

$$\mathrm{tr}\big[\Lambda J \Lambda J^\top\big] = \sum_{ik} \eta_i \eta_k \iint \tilde{v}_i(t) v_k(t) \, v_k(s) \, \tilde{v}_i(s) \, \mathrm{d}t \, \mathrm{d}s \,. \tag{56}$$

Here the corresponding contribution is

$$\mathrm{tr}\big[\Lambda J \Lambda J^\top\big] = \sum_i \eta_i \iint \tilde{v}_i(t) \sum_k \eta_k \langle v_k(t) \, v_k(s) \rangle \, \tilde{v}_i(s) \, \mathrm{d}t \, \mathrm{d}s \,, \tag{57}$$

where the summed variance $\sum_k \eta_k \langle v_k(t) \, v_k(s) \rangle$ of all modes causes a contribution to the variance of mode $i$ (due to the appearance of two $\tilde{v}_i$). The latter term is hence of order $\mathcal{O}(N)$ compared to the contribution (55); this shows that cross correlations between modes are smaller by an order $\mathcal{O}(N)$ compared to the variance terms, so that in the limit $N \to \infty$ the latter dominate.

We can also drop all terms where more than two blocks of $J$ and $J^\top$ appear, such as for example $\langle \mathrm{tr} \, J J^\top J J^\top \rangle = C \tilde{C} C \tilde{C} = \mathcal{O}(\tilde{C}^2)$, which is a term that will cancel because one $\tilde{C}$ corresponds to the appearance of a pair of fields $\tilde{v} \, \tilde{v}$ (causing an effective Gaussian noise), but the second $\tilde{C}$ would correspond to the appearance of a second pair $\langle \tilde{v} \tilde{v} \rangle \equiv 0$, which vanishes by the Ito convention (see (Coolen, 2001; Helias & Dahmen, 2020)). An analogous consideration holds for any order of perturbation theory and therefore also for the resummation of all such terms. In the VGA, this property is included because after differentiation by $\tilde{C}$ (in the equation of state (50)) one needs to set all remaining $\tilde{C}$ to zero due to the Ito convention. In the same manner, we need to drop all terms with two or more factors of $\bar{\tilde{v}}$, because we need to set $\bar{\tilde{v}} \equiv \langle \tilde{v} \rangle \equiv 0$ after differentiation, again by the Ito convention.

In summary, all leading order terms in the limit $N \to \infty$ contain concentrating quantities $C$ or $R$, respectively. Dropping the sub-leading terms (such as (56) as compared to (57)), the remaining action contains at most pairs of fields $v$ and $\tilde{v}$ and hence describes a Gaussian action. Moreover, the sub-leading terms are also the ones that would couple different modes $i \neq k$.

This means that the Gaussian action becomes one in which modes become independent in the limit (conditioned on the value of the concentrating fields $C$ and $R$). This together shows that the VGA in the $N \to \infty$ limit indeed becomes exact.

The terms in (52) are all cyclically invariant under the trace. This means that a term with $n$ factors in total and all factors identical ($J$ or $J^\top$) has a combinatorial factor of 1 as it appears exactly once when multiplying out (44)

$$\frac{1}{n} \operatorname{tr} \left[ \Lambda (J + J^\top) \right]^n = \frac{1}{n} \operatorname{tr} \left[ \Lambda (J + J^\top) \right] \ \ldots \ \left[ \Lambda (J + J^\top) \right].$$

By the cyclic invariance of the trace, a term that has $n - k$ factors of one sort ($J$ or $J^\top$) and $k$ factors of the respective other, appears $n$ times when multiplying out, because the position where the change from one factor to the other takes place can appear at $n$ positions; the respective change back is then fixed, because $k$ is fixed. In addition, each term with $n$ factors $J$ or $J^\top$ comes with a factor $1/n$.

So in total, the disorder term, in expectation under the Gaussian, takes the form

$$-\frac{1}{2} \left\langle \ln \det \left( \mathbb{I} - \Lambda (J + J^\top) \right) \right\rangle_{(v,\tilde{v}) \sim \mathcal{N}((\bar{v},\bar{\tilde{v}}),G)} \tag{58}$$

$$= \iint \bar{R}(t,s) \, dt \, ds$$

$$+ \frac{1}{2} \iiiint \bar{R}(t,u) \, \bar{R}(u,s) \, dt \, du \, ds + \frac{1}{2} \iiiint \bar{C}(t,u) \, \bar{\tilde{C}}(u,s) \, dt \, du \, ds$$

$$+ \sum_{n=3}^{\infty} \frac{1}{n} \int \left[ \bar{R} * \right]^n (t,t) \, dt$$

$$+ \frac{1}{2} \sum_{n=3}^{\infty} \sum_{k=1}^{n-1} \int \left[ \bar{R} * \right]^{n-k-1} \bar{C} \left[ \bar{R}^\top * \right]^{k-1} \bar{\tilde{C}}(\circ,t) \, dt \, ,$$

where we write $\left[ \bar{R} * \right]_{ts}^k = \int \cdots \int \bar{R}(t,u_1) \cdots \bar{R}(u_{k-1},s) \, du_1 \ldots du_{k-1}$.

We may now resum the terms that only depend on the response function as $\sum_{n=1}^{\infty} \frac{1}{n} \int \left[ \bar{R} * \right]^n dt = -\int \ln \left( 1 - \bar{R} \right) dt$. Likewise we may combine the terms that contain up to a single factor $\tilde{C}$ to obtain

$$-\frac{1}{2} \left\langle \ln \det \left( \mathbb{I} - \Lambda (J + J^\top) \right) \right\rangle_{(v,\tilde{v}) \sim \mathcal{N}\left[ (\bar{v},\bar{\tilde{v}}),G \right]} \tag{59}$$

$$= \sum_{n=1}^{\infty} \frac{1}{n} \int \left[ \bar{R} * \right]^n (t,t) \, dt$$

$$+ \frac{1}{2} \sum_{n=2}^{\infty} \sum_{k=1}^{n-1} \int \left[ \bar{R} * \right]^{n-k-1} \bar{C} \left[ \bar{R}^\top * \right]^{k-1} \bar{\tilde{C}}(\circ,t) \, dt \, .$$

## C.1. Equation of motion for the mean

Using (49) for $\delta/\delta\bar{\tilde{v}}$ we get

$$0 \stackrel{!}{=} \frac{\delta}{\delta\bar{\tilde{v}}} \left[ \left\langle \bar{S}(v,\tilde{v}) \right\rangle_{(v,\tilde{v}) \sim \mathcal{N}\left[ (\bar{v},\bar{\tilde{v}}),G \right]} \right] \Bigg|_{G^{(\tilde{v}\tilde{v})}=0, \tilde{v}=0} \, .$$

To evaluate this expression, we first compute the expectation value of the Gaussian part

$$\left\langle S_0 \right\rangle_{(v,\tilde{v}) \sim \mathcal{N}\left[ (\bar{v},\bar{\tilde{v}}),G \right]} \tag{60}$$

$$= \sum_{i=1}^{N} \iint \delta(t-s) \left\{ \left[ \partial_t + \frac{1}{g\beta} \right] \left[ \bar{\tilde{v}}_i(s)\bar{v}_i(t) + G_i^{(v\tilde{v})}(t,s) \right] + \beta^{-1} \left[ \bar{\tilde{v}}_i(t)\bar{\tilde{v}}_i(t) + G_i^{(\tilde{v}\tilde{v})}(t,s) \right] \right\} dt \, ds$$

$$- \int \frac{\bar{w}_i}{g\beta} \bar{\tilde{v}}_i(t) \, dt \, .$$

The contribution to the equation of state (49) is hence

$$\frac{\delta}{\delta\bar{\tilde{v}}_i(t)} \langle S_0 \rangle_{(v,\tilde{v})\sim\mathcal{N}\left[(\bar{v},\bar{\tilde{v}}),G\right]}$$

$$= \left[\partial_t + \frac{1}{g\beta}\right] \bar{v}_i(t) + \underbrace{G_i^{(v\tilde{v})}(t,t)}_{=0} - \frac{\bar{w}_i}{g\beta},$$

where $G_i^{(v\tilde{v})}(t,t) = 0$ by the Ito convention. The interaction term with (59) yields

$$\frac{\delta}{\delta\bar{\tilde{v}}_i(t)} \left\{ -\frac{P}{2} \left\langle \ln\det\left(\mathbb{I} - \Lambda(J+J^\top)\right)\right\rangle_{(v,\tilde{v})\sim\mathcal{N}\left[(\bar{v},\bar{\tilde{v}}),G\right]} \right\} \Bigg|_{G^{(\tilde{v}\tilde{v})}=0,\bar{\tilde{v}}=0} = P\eta_i \int \sum_{n=0}^{\infty} \left[\bar{R}*\right]^n(t,s)\, \bar{v}_i(s)\, ds\,,$$

where the factors $\eta_i$ and $\bar{v}_i(s)$ come from the inner derivative $\delta\bar{R}(s,t)/\delta\bar{\tilde{v}}(t)$ from (53) and all terms vanish where at least one factor $\tilde{C}$ or $\bar{\tilde{v}}$ remains after differentiation. So together we get the equation of motion for the mean

$$0 = \left[\partial_t + \frac{1}{g\beta}\right] \bar{v}_i(t) - \frac{\bar{w}_i}{g\beta} + P\eta_i \int \sum_{n=0}^{\infty} \left[\bar{R}*\right]^n(t,s)\, \bar{v}_i(s)\, ds. \tag{61}$$

### C.2. Solution of the response function

The Gaussian part $S_0$ of (42) in the equation of state takes the form (60). Because in the equation of state we set $\tilde{C} = 0$ ultimately, all terms containing $\tilde{C}$ drop out for the equation of state (42) for $G$; likewise, all terms $\propto \bar{\tilde{v}}$ vanish. So one has

$$\frac{\delta\langle S_0\rangle_{(v,\tilde{v})\sim\mathcal{N}(0,G)}}{\delta G_i^{(v\tilde{v})}(t,s)} \Bigg|_{G^{(\tilde{v}\tilde{v})}=0} = \delta(t-s)\left[\partial_t + \frac{1}{g\beta}\right].$$

The interaction term with (59) yields

$$\frac{\delta}{\delta G_i^{(v\tilde{v})}(t,s)} \left\{ -\frac{P}{2} \left\langle \ln\det\left(\mathbb{I} - \Lambda(J+J^\top)\right)\right\rangle_{(v,\tilde{v})\sim\mathcal{N}\left[(\bar{v},\bar{\tilde{v}}),G\right]} \right\} \Bigg|_{G^{(\tilde{v}\tilde{v})}=0} = P\eta_i \sum_{n=0}^{\infty} \left[\bar{R}*\right]^n(t,s)\,,$$

where the factor $\eta_i$ comes from the inner derivative $\delta\bar{R}/\delta G_i^{(v\tilde{v})}$ from (53). We also note that the term $\bar{w}$ drops out because we differentiated by $G^{(v\tilde{v})}$.

This yields the equation of state (49)

$$0 \overset{!}{=} \delta(t-s)\left[\partial_t + \frac{1}{g\beta}\right] + P\eta_i \sum_{n=0}^{\infty} \left[\bar{R}*\right]^n(t,s) + \left[G_i^{(\tilde{v}v)}\right]^{-1}(t,s)\,, \tag{62}$$

which is written as an implicit equation as

$$\left[\partial_t + \frac{1}{g\beta}\right] G_i^{(v\tilde{v})}(t,s) + P\eta_i \int \sum_{n=0}^{\infty} \left[\bar{R}*\right]^n(t,u)\, G_i^{(v\tilde{v})}(u,s)\, \mathrm{d}u = -\delta(t-s)\,. \tag{63}$$

Moving to Fourier domain and resumming the geometric series one has

$$\left[i\omega + \frac{1}{g\beta}\right] G_i^{(v\tilde{v})}(\omega) + P\eta_i \frac{1}{1-\bar{R}(\omega)} G_i^{(v\tilde{v})}(\omega) = -1\,. \tag{64}$$

We notice that the self-feedback mediated by the response functions is an alternating series, since $G_i^{(v\tilde{v})}(t,s) < 0$. The latter equation needs to be solved self-consistently together with the definition of the response function $\bar{R}$ by (53) at the saddle point $\bar{\tilde{v}} = 0$

$$\bar{R}(t,s) = \sum_{i=1}^{N} \eta_i G_i^{(v\tilde{v})}(t,s)\,.$$

Likewise, we formally get an equation of motion for $\bar{\tilde{v}}(t)$ by differentiating by $\bar{v}(t)$, which indeed admits the solution $\bar{\tilde{v}}(t) = 0$, since the resulting equation contains at least one factor of $\bar{\tilde{v}}$ in each term.

### C.3. Solution of the correlation function

The equation of state with regard to $G_i^{(\tilde{v}\tilde{v})}$ yields the self-consistency equation for the autocorrelation. The Gaussian part yields (60)

$$\frac{\delta \langle S_0 \rangle_{(v,\tilde{v})\sim\mathcal{N}\left[(\bar{v},\bar{\tilde{v}}),G\right]}}{\delta G_i^{(\tilde{v}\tilde{v})}(t,s)}\Bigg|_{G^{(\tilde{v}\tilde{v})}=0,\bar{\tilde{v}}=0} = \beta^{-1}\,\delta(t-s)\,,$$

and the term (59) yields

$$\frac{\delta}{\delta G_i^{(\tilde{v}\tilde{v})}(t,s)}\left\{-\frac{P}{2}\left\langle \ln\det\left(\mathbb{I}-\Lambda(J+J^\top)\right)\right\rangle_{(v,\tilde{v})\sim\mathcal{N}\left[(\bar{v},\bar{\tilde{v}}),G\right]}\right\}\Bigg|_{G^{(\tilde{v}\tilde{v})}=0,\bar{\tilde{v}}=0}$$

$$= \frac{P}{2}\eta_i \sum_{n=2}^{\infty}\sum_{k=1}^{n-1}\left\{\left[\bar{R}*\right]^{n-k-1}\bar{C}\left[\bar{R}^\top *\right]^{k-1}\right\}(t,s)\,,$$

so that together we have the equation of state (50)

$$0 = \beta^{-1}\,\delta(t-s) + \frac{P}{2}\eta_i \sum_{n=2}^{\infty}\sum_{k=1}^{n-1}\left\{\left[\bar{R}*\right]^{n-k-1}\bar{C}\left[\bar{R}^\top *\right]^{k-1}\right\}(t,s) - \frac{1}{2}\left[G^{(v\tilde{v})}\right]^{-1}G^{(vv)}\left[G^{(\tilde{v}v)}\right]^{-1}. \tag{65}$$

We may insert (62) to get the equation of motion for $G^{(vv)}$ as

$$\iint\left[\left(\partial_t + \frac{1}{g\beta}\right)\delta(t-t') + P\eta_i\sum_{n=0}^{\infty}\left[\bar{R}*\right]^n(t,t')\right] \tag{66}$$

$$\times\left[\left(\partial_s + \frac{1}{g\beta}\right)\delta(s-s') + P\eta_i\sum_{n=0}^{\infty}\left[\bar{R}*\right]^n(s,s')\right]G_i^{(vv)}(t',s')\,\mathrm{d}t'\,\mathrm{d}s'$$

$$= 2\beta^{-1}\,\delta(t-s) + P\eta\sum_{n=2}^{\infty}\sum_{k=1}^{n-1}\left\{\left[\bar{R}*\right]^{n-k-1}\bar{C}\left[\bar{R}^\top *\right]^{k-1}\right\}(t,s)\,.$$

## D. Effective equation of motion

We may interpret the equation of motion for the response function (63) together with the equation of state for the correlation function (66) as both resulting from the same the effective stochastic equation of motion

$$\left[\partial_t + \frac{1}{g\beta}\right]\delta v_i(t) + P\eta_i\int \sum_{n=0}^{\infty}\left[\bar{R}*\right]^n(t,s)\,\delta v_i(s)\,\mathrm{d}s = \xi_i(t)\,. \tag{67}$$

The mean, in addition, follows the equation (61), which may be included here to yield the effective equation given in the main text (20).

The resummed response function $K(t,s) = \sum_{n=0}^{\infty}\left[\bar{R}*\right]^n(t,s)$ may be regarded at the solution $K$ of the implicit equation

$$K(t,s) = \int \mathrm{d}u\,\bar{R}(t-u)\,K(u-s) + \delta(t-s) \tag{68}$$

so that one has

$$\left[\partial_t + \frac{1}{g\beta}\right]\delta v_i(t) + P\eta_i\int K(t,s)\,\delta v_i(s)\,\mathrm{d}s = \xi_i(t)\,, \tag{69}$$

where the variance of the noise, according to the right hand side of (66), is given by

$$\langle \xi_i(t)\xi_j(s)\rangle = \delta_{ij}\left[2\beta^{-1}\,\delta(t-s) + P\eta_i\sum_{n=2}^{\infty}\sum_{k=1}^{n-1}\left\{\left[\bar{R}*\right]^{n-k-1}\bar{C}\left[\bar{R}^\top *\right]^{k-1}\right\}(t,s)\right]. \tag{70}$$

The double sum may be rewritten as

$$
\sum_{n=2}^{\infty} \sum_{k=1}^{n-1} [\dots]^{n-k-1} [\dots]^{k-1}
$$

$$
= \sum_{n=0}^{\infty} \sum_{k=0}^{n} [\dots]^{n-k} [\dots]^{k}
$$

$$
= \sum_{l=0}^{\infty} \sum_{k=0}^{\infty} [\dots]^{l} [\dots]^{k} ,
$$

by renaming $n - k =: l$, so that we may write the correlator (70) with (68) as

$$
\langle \xi_i(t) \xi_j(s) \rangle = \delta_{ij} \left[ 2\beta^{-1} \delta(t-s) + P\eta_i \left\{ K * \bar{C} * K^\top \right\}(t,s) \right] \quad . \tag{71}
$$

## E. Generalization Error

### E.1. Variance part

Writing the effective equation of motion of the discrepancy 67 as

$$
L[\delta v] = \xi
$$

where $L$ is a linear operator and $\xi$ is an inhomogeneity, the solution can be written in terms of the Green's function as

$$
\delta v_i(t) = \int \mathrm{d}s \, G_{ij}(t-s) \xi_j(s)
$$

with

$$
L[G] = \delta_{ij} \delta(t-s) \quad .
$$

The Green's function can be interpreted as the $\langle v\tilde{v} \rangle$-correlator in 25 and the variance can be obtained by taking the average utilizing (70)

$$
\langle \delta v_i(t) \delta v_j(s) \rangle = \iint \mathrm{d}u \mathrm{d}u' G_{ik}(t-u) \langle \xi_k(u) \xi_l(u') \rangle G_{jl}(s-u')
$$

$$
= 2\beta^{-1} \int_0^{\min(t,s)} \mathrm{d}u G_i(t-u) G_j(u-s) \delta_{ij}
$$

$$
+ P\eta_i \int_0^t \int_0^s \mathrm{d}u \mathrm{d}u' G_i(t-u) \left\{ K * \bar{C} * K^\top \right\}(u,u') G_j(s-u') \delta_{ij} \quad .
$$

### E.2. Bias part

As for the variance part, the mean $\bar{v}$ obeys the same differential equation as $\delta v$

$$
L[\bar{v}] = \frac{\bar{w}}{g\beta} \quad .
$$

Hence, the solution takes on the form

$$
\bar{v} = \bar{v}_{\mathrm{hom}} + \bar{v}_{\mathrm{part}}
$$

$$
= \bar{v}_{\mathrm{hom}} + \frac{\bar{w}}{g\beta} \int_0^t \mathrm{d}s \, G(t-s) \quad .
$$

While for $\delta v$ the homogeneous solution that satisfies the boundary condition is $\delta v \equiv 0$ by design, the boundary conditions for $\bar{v}$ cannot be satisfied with a vanishing homogeneous solution, if the system starts in a *tabula rasa* state, i.e. $\bar{v}(0) = \bar{w}$. In the stationary state the homogeneous parts of the fields have to vanish as before, i.e. $\bar{v}_{\mathrm{hom}}(\infty) = 0$. As the Green's function satisfies the homogeneous equation, the full solution is found to be

$$
\bar{v}_i(t) = \bar{w}_i G_i(t) + \frac{\bar{w}_i}{g\beta} \int_0^t \mathrm{d}s \, G(t-s) \quad .
$$

### E.3. Generalization error

The generalization error is given with (31) as

$$\mathcal{L}_{\text{test}}(t) = \frac{1}{2}\bar{C}(t,t)$$
$$= \frac{1}{2}\sum_i \eta_i \left\langle \delta v_i(t)^2 \right\rangle + \frac{1}{2}\sum_i \eta_i \bar{v}_i(t)^2 \quad .$$

## F. Bayesian interpretation of the partition function

Writing the thermodynamic partition function as

$$\mathcal{Z} = \int \mathrm{d}w \, e^{-\beta H(w,\mathcal{D})}$$
$$= \int \mathrm{d}w \, e^{-\frac{\beta}{2}\sum_{\alpha=1}^{P}(y_\alpha - f_\alpha)^2 - \frac{1}{2g}\sum_{i=1}^{N} w_i^2}$$

with the Hamiltonian $H$ defined in (3) allows for a probabilistic viewpoint of the partition function as

$$\mathcal{Z} \propto \int \mathrm{d}w \, \mathcal{N}(y|f(w),\beta^{-1})\,\mathcal{N}(w|0,g)\,.$$

Because $f$ is a linear function of $w$ its distribution is again Gaussian with variance

$$\left\langle f(x_\mu)f(x_\nu)\right\rangle_{p(w)} = \left\langle w^\top \psi(x_\mu)\psi(x_\nu)^\top w\right\rangle_{p(w)} = g\sum_i \psi_i(x_\mu)\psi_i(x_\nu)\,,$$

where the latter can be identified as the kernel $\kappa(x_\mu, x_\nu)$ 6. From here it follows

$$p(y) = \mathcal{N}(y|0,\kappa + \beta^{-1}\mathbb{I})\,,$$

where the Langevin noise $\beta^{-1}$ acts as a regulator to the distribution of the $y$ akin to a static label noise.

## G. Derivation of Scaling Laws

We obtain a natural lower bound on the generalization error by considering the limit in which we neglect the disorder of the matrix $Q$ in (14), replacing $Q \to \langle Q \rangle$, so that the mutual slowing down between individual nodes vanishes. Effectively, this treats the weights as independent Ornstein-Uhlenbeck like processes. In this limit we have

$$K(t-s) = \delta(t-s)$$

and

$$G_i(t-s) = \Theta(t-s)e^{-\left(P\eta_i + \frac{1}{g\beta}\right)(t-s)}$$

with $\Theta$ the Heaviside function following the convention $\Theta(0) = 0$.

### G.1. Bias part

The bias part $\mathcal{L}_{\text{bias}}$ of the loss simplifies to $(t \gg 1)$

$$\mathcal{L}_{\text{bias}}(t) = \frac{1}{2}\sum_i \eta_i \bar{w}_i^2 e^{-2\left(P\eta_i + \frac{1}{g\beta}\right)t} \quad .$$

A power-law arises in the case where regularization is small, i.e. $g\beta P\eta_i \gg 1$. After averaging over $\bar{w}_i \sim \mathcal{N}(0,1)$, in the large $N$ limit, the sum can be treated as an integral

$$
\begin{aligned}
\langle \mathcal{L}_{\text{bias}}(t) \rangle_{\bar{w}} &\approx \frac{1}{2} \sum_{i=1}^{N} \eta_i e^{-2P\eta_i t} \\
&= \frac{1}{2} \sum_{i=1}^{N} i^{-1-\alpha} e^{-2Pi^{-1-\alpha}t} \\
&\approx \frac{1}{2} \int_1^N i^{-1-\alpha} e^{-2Pi^{-1-\alpha}t} \mathrm{d}i \\
&= \frac{1}{2} \frac{(2Pt)^{-\alpha/(1+\alpha)}}{1+\alpha} \int_{2Pt/N^{1+\alpha}}^{2Pt} u^{-1/(1+\alpha)} e^{-u} \, \mathrm{d}u \\
&\overset{Pt \gg 1}{=} \frac{1}{2} \frac{(2Pt)^{-\alpha/(1+\alpha)}}{1+\alpha} \left( \Gamma\left(\frac{\alpha}{1+\alpha}\right) - \gamma\left(\frac{\alpha}{1+\alpha}, 2Pt/N^{1+\alpha}\right) \right) \\
&= \frac{1}{2} \frac{(2Pt)^{-\alpha/(1+\alpha)}}{1+\alpha} \Gamma\left(\frac{\alpha}{1+\alpha}\right) - \frac{1}{2} N^{-\alpha}/\alpha + \mathcal{O}\left(N^{-\alpha}(Pt)\right),
\end{aligned}
$$

where $\gamma(a, b)$ is the lower incomplete Gamma function (DLM). Note that this only works for $\alpha > 0$. This limit yields a similar result as eq. 125 in (Bordelon et al., 2024).

## G.2. Variance part

The variance part of the generalization error $\mathcal{L}_{\text{var}}$ also shows power-law behavior under certain assumptions. If we want to investigate early stopping in the regime of high temperature, the main contribution comes from the $\beta$-dependent part of $\mathcal{L}_{\text{var}}$

$$
\begin{aligned}
\mathcal{L}_{\text{var}}(t) \approx \mathcal{L}_\beta(t) &:= \frac{1}{\beta} \sum_{i=1}^{N} \eta_i \int_0^t \mathrm{d}s \, G_i(t-s)^2 \\
&\approx \frac{1}{\beta} \sum_{i=1}^{N} \eta_i \int_0^t \mathrm{d}s \, e^{-2P\eta_i(t-s)} \\
&= \frac{1}{2P\beta} \sum_{i=1}^{N} \left[ 1 - e^{-2P\eta_i t} \right]
\end{aligned}
$$

In the same manner as before, this can rewritten as

$$
\begin{aligned}
2P\beta \mathcal{L}_\beta(t) &= N - \sum_{i=1}^{N} e^{-2P\eta_i t} \\
&\approx N - \int_1^N e^{-2Pi^{-1-\alpha}t} \mathrm{d}i \\
&= N - \frac{(2Pt)^{1/(1+\alpha)}}{1+\alpha} \int_{2Pt/N^{1+\alpha}}^{2Pt} u^{-1/(1+\alpha)-1} e^{-u} \, \mathrm{d}u \\
&= N - \frac{(2Pt)^{1/(1+\alpha)}}{1+\alpha} \left( \Gamma\left(-\frac{1}{1+\alpha}\right) - \gamma\left(-\frac{1}{1+\alpha}, \frac{2Pt}{N^{1+\alpha}}\right) \right) \\
&\approx N - \frac{(2Pt)^{1/(1+\alpha)}}{1+\alpha} \Gamma\left(-\frac{1}{1+\alpha}\right) + N + \mathcal{O}\left(N^{-\alpha}(Pt)\right) \\
&= (2Pt)^{1/(1+\alpha)} \Gamma\left(\frac{\alpha}{1+\alpha}\right) + \mathcal{O}\left(N^{-\alpha}(Pt)\right) \quad .
\end{aligned}
$$

### G.3. Optimal generalization error

The optimal stopping point can be estimated as the stationary point of the test error that satisfies

$$
\begin{aligned}
0 &= \left. \frac{\mathrm{d}\left(\mathcal{L}_{\mathrm{bias}}(t) + \mathcal{L}_\beta(t)\right)}{\mathrm{d}t} \right|_{t=t^*} \\
&= \left. \frac{\mathrm{d}}{\mathrm{d}t} \left( \frac{(2Pt)^{-\alpha/(1+\alpha)}}{1+\alpha} + \frac{(2Pt)^{1/(1+\alpha)}}{P\beta} \right) \right|_{t=t^*} \\
&= \left. \frac{\mathrm{d}}{\mathrm{d}t} \left( \frac{(2P)^{1/(1+\alpha)} t^{-\alpha/(1+\alpha)}}{2P(1+\alpha)} + \frac{(2Pt)^{1/(1+\alpha)}}{P\beta} \right) \right|_{t=t^*} \\
&= -\frac{\alpha(t^*)^{-\alpha/(1+\alpha)-1}}{2(1+\alpha)^2} + \frac{1}{\beta}\frac{(t^*)^{-\alpha/(1+\alpha)}}{1+\alpha} \\
\Leftrightarrow t^* &= \frac{\beta}{2}\frac{\alpha}{(1+\alpha)} \quad .
\end{aligned}
$$

Evaluating $\mathcal{L}_{\text{test}}$ at this point yields

$$
\begin{aligned}
\mathcal{L}_{\text{test}}(t^*) &\approx \Gamma\left(\frac{\alpha}{1+\alpha}\right)\left(\frac{(2Pt^*)^{1/(1+\alpha)}}{2P\beta} + \frac{1}{2}\frac{(2Pt^*)^{-\alpha/(1+\alpha)}}{1+\alpha}\right) - \frac{1}{2\alpha}N^{-\alpha} \\
&= \frac{1}{2}(P\beta)^{-\alpha/(1+\alpha)}\Gamma\left(\frac{\alpha}{1+\alpha}\right)\left(\left(\frac{\alpha}{(1+\alpha)}\right)^{1/(1+\alpha)} + \frac{\left(\frac{\alpha}{(1+\alpha)}\right)^{-\alpha/(1+\alpha)}}{1+\alpha}\right) - \frac{1}{2\alpha}N^{-\alpha} \\
&= \frac{1}{2}\Gamma\left(\frac{\alpha}{1+\alpha}\right)\left(\frac{P\beta\alpha}{(1+\alpha)}\right)^{-\alpha/(1+\alpha)} - \frac{1}{2\alpha}N^{-\alpha} \\
&= \frac{1}{2}\Gamma\left(\frac{\alpha}{1+\alpha}\right)(2Pt^*)^{-\alpha/(1+\alpha)} - \frac{1}{2\alpha}N^{-\alpha}
\end{aligned}
$$

We first note, that this approximation can work when the spectrum decays faster than $1/i$ which is in agreement with the fact that for a slower decay $\mathcal{L}_{\mathrm{bias}}$ and $\mathcal{L}_\beta$ would become extensive in the number of parameters. The optimal stopping time is the later the faster the spectrum decays corresponding to the system being able to improve generalization for longer during training and minimizing $\mathcal{L}_{\text{test}}$ at a lower value. It is however limited by the inverse temperature which induces an upper threshold for $t^*$ and thereby an implicit timescale on the system as the time where the bias has decayed such that the accumulated fluctuations are of the same order of magnitude. Surprisingly, the optimal stopping point is independent of the number of parameters as well as the number of samples to this order of approximation.

On the level of the generalization error we recover the intuitions that more samples as well as more parameters aid generalization. The sample size $P$ appears as a conjugate to the inverse temperature $\beta$ in the way that in this regime noise and sample scale exactly anti-proportional.

Our results agree with (Bordelon et al., 2024) in the case that $t$ and $N$ are large with the corresponding other parameters go to infinity but differ in the case of finite $P$ while $t, N \to \infty$ as we are not able to take the $t \to \infty$ limit easily. This is due to the fact, that we consider the $\beta$-dependent part as the dominating contribution while it is not present in their work and hence the other contributions to the loss become dominant. In the noiseless case we can take the stationary limit easily and recover results from the literature as discussed in Section I.

## H. Disorder average with label noise

To include a read-out noise on the labels (2), i.e.

$$
y^\mu = \bar{w}^\top \psi(x^\mu) + \epsilon^\mu
$$

with $\epsilon^\mu \sim \mathcal{N}(0, \sigma^2)$ the Loss function $H$ (3) takes the form

$$H := \frac{1}{2} \sum_{\mu=1}^{P} \left[ \sum_{i=1}^{N} (\overline{w}_i - w_i) \psi_i(x_\mu) - \epsilon_\mu \right]^2 + \frac{1}{2g\beta} \sum_{i=1}^{N} w_i^2$$

$$= \frac{1}{2} \sum_{i,j=1}^{N} (\overline{w}_i - w_i) \sum_\mu \psi_i(x_\mu) \psi_j(x_\mu) (\overline{w}_j - w_j)$$

$$- \sum_{i=1}^{N} (\overline{w}_i - w_i) \sum_{\mu=1}^{P} \psi_i(x_\mu) \epsilon_\mu + \frac{1}{2g\beta} \sum_{i=1}^{N} w_i^2 + \text{const.}$$

which leads to the equation of motion

$$\frac{\partial}{\partial t} w_i(t) = \sum_{j=1}^{N} \sum_{\mu=1}^{P} \psi_i(x_\mu) \psi_j(x_\mu) (\overline{w}_j - w_j) - \frac{1}{g\beta} w_i + \sum_{\mu=1}^{P} \psi_i(x_\mu) \epsilon_\mu + \zeta_i(t) \,.$$

So the new action (39) becomes with $v = \bar{w} - w$

$$S(v, \tilde{v}) = \int dt \sum_{i=1}^{N} \tilde{v}_i(t) \left( \partial_t v_i(t) + \sum_{j=1}^{N} \sum_{\mu=1}^{P} \psi_i(x_\mu) \psi_j(x_\mu) v_j(t) + \frac{v_i - \overline{w}_i}{g\beta} + \sum_{\mu=1}^{P} \psi_i(x_\mu) \epsilon_\mu \right) + \frac{1}{\beta} \sum_{i=1}^{N} \tilde{v}_i(t)^2 \,.$$

The term $\sum_{\mu=1}^{P} \psi_i(x_\mu) \epsilon_\mu$ is linear in $\epsilon$, therefore, we can average the moment generating function $Z$ directly over the label noise and obtain for the action

$$S(v, \tilde{v}) = \int dt \sum_{i=1}^{N} \tilde{v}_i(t) \left( \partial_t v_i(t) + \sum_{j=1}^{N} \sum_{\mu=1}^{P} \psi_i(x_\mu) \psi_j(x_\mu) v_j(t) + \frac{v_i - \overline{w}_i}{g\beta} \right) + \frac{1}{\beta} \sum_{i=1}^{N} \tilde{v}_i(t)^2$$

$$+ \frac{\sigma^2}{2} \sum_{i=1}^{N} \iint \tilde{v}_i(t) \tilde{v}_j(s) \, dt \, ds \sum_{\mu=1}^{P} \psi_i(x_\mu) \psi_j(x_\mu) \,.$$

Structurally, this appears as an additive factor to the Langevin noise of the OUP, i.e.

$$\frac{2}{\beta} \delta_{ij} \delta(t - s) \rightarrow \frac{2}{\beta} \delta_{ij} \delta(t - s) + \sigma^2 \sum_{\mu=1}^{P} \psi_i(x_\mu) \psi_j(x_\mu)$$

introducing a frozen noise term coupling both parameters and time points. Taking the disorder average over the $\psi$ akin to the derivation of (A), we get

$$\bar{S}(v, \tilde{v}) = \int dt \sum \tilde{v}_i(t) \left( \left[ \partial_t + \frac{1}{g\beta} \right] v_i(t) - \frac{1}{g\beta} \overline{w}_i \right) + \frac{1}{\beta} \sum_{i=1}^{N} \tilde{v}_i(t)^2$$

$$- \frac{P}{2} \ln \det \left( \mathbb{I} - \Lambda \left[ \int \tilde{v}(t) v(t)^\top + v(t) \tilde{v}(t)^\top dt + \sigma^2 \iint \tilde{v}(t) \tilde{v}(s)^\top dt \, ds \right] \right) \,. \tag{72}$$

### H.1. Identification of order parameters

Similar to (B) we introduce $J := \int \tilde{v}(t) v(t)^\top \, dt$ and $M := \sigma^2 \iint \tilde{v}(t) \tilde{v}(s)^\top \, dt \, ds$. Note that each factor $J$ comes with a single time integral, while each factor $M$ comes with two time integrals. We expand the $\ln \det$ with help of the logarithm series as

$$-\frac{1}{2} \ln \det \left( \mathbb{I} - \Lambda \left( J + J^\top + M \right) \right) \tag{73}$$

$$= \frac{1}{2} \sum_{n=1}^{\infty} \frac{1}{n} \operatorname{tr} \left[ \Lambda \left( J + J^\top + M \right) \right]^n \tag{74}$$

$$= \frac{1}{2} \sum_i \eta_i J_{ii} + \sum_i \eta_i J_{ii}^\top + \sum_i \eta_i M_{ii}$$

$$+ \frac{1}{4} \sum_i \sum_j \eta_i J_{ij} \, \eta_j J_{ji} + \frac{1}{4} \sum_i \sum_j \eta_i J_{ij} \, \eta_j J_{ji}^\top + \frac{1}{4} \sum_i \sum_j \eta_i J_{ij}^\top \, \eta_j J_{ji} + \frac{1}{4} \sum_i \sum_i \eta_i j_{ij}^\top \, \eta_j j_{ji}^\top$$

$$+ \frac{1}{4} \sum_i \sum_j \eta_i J_{ij} \, \eta_j M_{ji} + \frac{1}{4} \sum_i \sum_j \eta_i J_{ij}^\top \, \eta_j M_{ji} + \frac{1}{4} \sum_i \sum_j \eta_i M_{ij} \, \eta_j J_{ji} + \frac{1}{4} \sum_i \sum_j \eta_i M_{ij} \, \eta_j J_{ji}^\top$$

$$+ \frac{1}{4} \sum_i \sum_j \eta_i M_{ij} \, \eta_j M_{ji} \tag{75}$$

$$+ \dots .$$

So the following products arise

$$\left[ \Lambda J \, \Lambda J \right]_{ij} \equiv \iint \eta_i \tilde{v}_i(t) \underbrace{\sum_k \eta_k v_k(t) \, \tilde{v}_k(s)}_{=: R(t,s)} v_j(s) \, \mathrm{d}t \, \mathrm{d}s \, , \tag{76}$$

$$\left[ \Lambda J^\top \Lambda J^\top \right]_{ij} \equiv \iint \eta_i v_i(t) \underbrace{\sum_k \eta_k \tilde{v}_k(t) \, v_k(s)}_{=: R(s,t)} \tilde{v}_j(s) \, \mathrm{d}t \, \mathrm{d}s \, ,$$

$$\left[ \Lambda J^\top \Lambda J \right]_{ij} \equiv \iint \eta_i v_i(t) \underbrace{\sum_k \eta_k \tilde{v}_k(t) \, \tilde{v}_k(s)}_{=: \tilde{C}(t,s)} v_j(s) \, \mathrm{d}t \, \mathrm{d}s \, ,$$

$$\left[ \Lambda J \, \Lambda J^\top \right]_{ij} \equiv \iint \eta_i \tilde{v}_i(t) \underbrace{\sum_k \eta_k v_k(t) \, v_k(s)}_{=: C(t,s)} \tilde{v}_j(s) \, \mathrm{d}t \, \mathrm{d}s \, ,$$

$$\left[ \Lambda J \, \Lambda M \right]_{ij} \equiv \iiint \eta_i \tilde{v}_i(t) \underbrace{\sum_k \eta_k v_k(t) \, \tilde{v}_k(s)}_{=: R^\top(t,s)} \sigma^2 \tilde{v}_j(u) \, \mathrm{d}t \, \mathrm{d}s \, \mathrm{d}u,$$

$$\left[ \Lambda J^\top \, \Lambda M \right]_{ij} \equiv \iiint \eta_i v_i(t) \underbrace{\sum_k \eta_k \tilde{v}_k(t) \, \tilde{v}_k(s)}_{=: \tilde{C}(t,s)} \sigma^2 \, \tilde{v}_j(u) \, \mathrm{d}t \, \mathrm{d}s \, \mathrm{d}u,$$

$$\left[ \Lambda M \, \Lambda J \right]_{ij} \equiv \iiint \eta_i \tilde{v}_i(t) \sigma^2 \underbrace{\sum_k \eta_k \tilde{v}_k(s) \, \tilde{v}_k(u)}_{=: \tilde{C}(s,u)} v_j(u) \, \mathrm{d}t \, \mathrm{d}s \, \mathrm{d}u,$$

$$\left[ \Lambda M \, \Lambda J^\top \right]_{ij} \equiv \iiint \eta_i \tilde{v}_i(t) \sigma^2 \underbrace{\sum_k \eta_k \tilde{v}_k(s) \, v_k(u)}_{=: R(u,s)} \tilde{v}_j(u) \, \mathrm{d}t \, \mathrm{d}s \, \mathrm{d}u,$$

$$\left[\Lambda M\,\Lambda M\right]_{ij} \equiv \iiiint \eta_i \tilde{v}_i(t)\sigma^2 \underbrace{\sum_k \eta_k \tilde{v}_k(s)\,\tilde{v}_k(u)}_{=:\tilde{C}(s,u)}\,\sigma^2 \tilde{v}_j(r)\,\mathrm{d}t\,\mathrm{d}s\,\mathrm{d}u\,\mathrm{d}r \quad .$$

In the following, we leave it implicit that a factor $J$ comes with a single time integral and a factor $M$ with two independent time integrals. With this shorthand, we obtain the rules

$$J\Lambda J \to \tilde{v}Rv,$$
$$J^\top \Lambda J^\top \to vR^\top \tilde{v},$$
$$J^\top \Lambda J \to v\tilde{C}v,$$
$$J\Lambda J^\top \to \tilde{v}C\tilde{v},$$
$$J\Lambda M \to \tilde{v}R^\top \tilde{v},$$
$$J^\top \Lambda M \to v\tilde{C}\tilde{v},$$
$$M\Lambda J \to \tilde{v}\tilde{C}v,$$
$$M\Lambda J^\top \to \tilde{v}R\tilde{v}$$
$$M\Lambda M \to \tilde{v}\tilde{C}\tilde{v}.$$

We may group these rules according to which fields remain to better see the structure

$$
\begin{aligned}
&1.\{JJ,MJ\} &&\to \tilde{v}\{R,\tilde{C}\}v,\\
&2.\{J^\top J^\top,J^\top M\} &&\to v\{R^\top,\tilde{C}\}\tilde{v},\\
&3.J^\top J &&\to v\tilde{C}v,\\
&4.\{JJ^\top,JM,MJ^\top,MM\} &&\to \tilde{v}\{C,R^\top,R,\tilde{C}\}\tilde{v}.
\end{aligned}
$$

In the VGA we only need to keep those terms where at most a single factor $\tilde{C}$ or $\tilde{v}$ appears, because in the equation of state we will differentiate by either of these variables once and subsequently set it to zero – any term containing more than a single power of $\tilde{C}$ or $\tilde{v}$ thus vanishes.

We may therefore

- drop terms with factors $MM$, because they yield one $\tilde{C}$ and two remaining $\tilde{v}$, which would cause another $\tilde{C}$ ultimately

- drop terms which include more than a single factor $J^\top J$

- terms that include a single factor $J^\top J$ already contain a single $\tilde{C}$, so they can only be combined from left with $J^\top J^\top$ and from right with $JJ$ (and not with $J^\top M$ or $MJ$, because this would yield another factor $\tilde{C}$)

- drop terms with more than a single occurrence of $MJ$ or $J^\top M$

- drop terms with more than a single occurrence from set 4. $\{JJ^\top,JM,MJ^\top\}$, because each such occurrence causes a pair of $\tilde{v}$ to remain on the left and on the right, which ultimately needs to be contracted to yield a $\tilde{C}$ on either side

- terms which include one element of set 4. $\{JJ^\top,JM,MJ^\top\}$ can only be combined with factors $JJ$ from left and $J^\top J^\top$ from the right from set 1. (and not with $MJ$ or $JM$), for otherwise there would be more than a single $\tilde{C}$ in the end

The remaining terms are

$$\langle \mathrm{tr}(\Lambda J)^n\rangle_{(v,\tilde{v})\sim\mathcal{N}\left[(\bar{v},\bar{\tilde{v}}),G\right]} \to \left[\bar{R}\right]^n,$$

$$\langle \mathrm{tr}(\Lambda J^\top)^n\rangle_{(v,\tilde{v})\sim\mathcal{N}\left[(\bar{v},\bar{\tilde{v}}),G\right]} \to \left[\bar{R}^\top\right]^n,$$

(77)

$$\langle \mathrm{tr}(\Lambda J)^{n-k}\,(\Lambda J^\top)^k \rangle_{(v,\tilde{v})\sim\mathcal{N}\left[(\bar{v},\bar{\tilde{v}}),G\right]} \overset{1\le k\le n-1}{\Rightarrow} \left[\bar{R}\right]^{n-k-1}\bar{C}\left[\bar{R}^\top\right]^{k-1}\bar{\tilde{C}},$$

$$\langle \mathrm{tr}(\Lambda J^\top)^{n-k}\,(\Lambda J)^k \rangle_{(v,\tilde{v})\sim\mathcal{N}\left[(\bar{v},\bar{\tilde{v}}),G\right]} \overset{1\le k\le n-1}{\Rightarrow} \left[\bar{R}^\top\right]^{n-k-1}\bar{\tilde{C}}\left[\bar{R}\right]^{k-1}\bar{C},$$

$$\langle \mathrm{tr}(\Lambda J)^{n-k}\,M\,(\Lambda J^\top)^{k-1} \rangle_{(v,\tilde{v})\sim\mathcal{N}\left[(\bar{v},\bar{\tilde{v}}),G\right]} \overset{1\le k\le n-1}{\Rightarrow} \left[\bar{R}\right]^{n-k-1}\bar{R}\left[\bar{R}^\top\right]^{k-2}\bar{\tilde{C}},$$

$$\langle \mathrm{tr}(\Lambda J^\top)^{n-k}\,M\,(\Lambda J^\top)^{k-1} \rangle_{(v,\tilde{v})\sim\mathcal{N}\left[(\bar{v},\bar{\tilde{v}}),G\right]} \to \left[\bar{R}^\top\right]^{n-k-1}\bar{\tilde{C}}\left[\bar{R}^\top\right]^{k},$$

$$\langle \mathrm{tr}(\Lambda J)^{n-k}\,M\,(\Lambda J)^{k-1} \rangle_{(v,\tilde{v})\sim\mathcal{N}\left[(\bar{v},\bar{\tilde{v}}),G\right]} \to \left[\bar{R}\right]^{n-k}\bar{\tilde{C}}\left[\bar{R}\right]^{k-1} \tag{78}$$

Due to the different time arguments in the $\sigma^2\iint \tilde{v}(t)\tilde{v}(s)\mathrm{d}t\mathrm{d}s$-term, we get additional integrals compared to (54). In particular for

$$\langle \mathrm{tr}(JMJ) \rangle = \sigma^2 \int \cdots \int \langle \tilde{v}(t)v(t)\tilde{v}(s)\tilde{v}(u)\tilde{v}(w)v(w) \rangle\, \mathrm{d}t\,\mathrm{d}s\,\mathrm{d}u\,\mathrm{d}w$$

$$= \sigma^2 \int \cdots \int \langle v(t)\tilde{v}(s) \rangle \langle \tilde{v}(u)\tilde{v}(w) \rangle \langle v(w)\tilde{v}(t) \rangle\, \mathrm{d}t\,\mathrm{d}s\,\mathrm{d}u\,\mathrm{d}w$$

$$\sim \sigma^2 \int \cdots \int R(t-s)\tilde{C}(u,w)R(w-t)\, \mathrm{d}t\,\mathrm{d}s\,\mathrm{d}u\,\mathrm{d}w$$

with all $\Lambda$ to be imagined in the right places. Collecting all higher orders and summing over them would yield a term (after differentiation w.r.t. $\tilde{C}(u,w)$)

$$\sigma^2 \int K^*(w-s)\,\mathrm{d}s$$

with $K^* \equiv \sum_{k=1}^\infty R^k = K - \delta$, where interestingly this term is $u$ independent and integrated over $s$. Similarly for the $\langle \mathrm{tr}(\Lambda J^\top)^{n-1}M \rangle_{(v,\tilde{v})\sim\mathcal{N}\left[(\bar{v},\bar{\tilde{v}}),G\right]}$ terms, one gets a factor of the form

$$\sigma^2 \int K^{*\top}(w-s)\,\mathrm{d}s\,.$$

The reason to introduce $K^*$ is that the zeroth order term ($\langle M \rangle = \sigma^2\tilde{C}$) can only appear in one of the three terms which is why we will keep it separate.

Finally, there are the mixed terms

$$\langle \mathrm{tr}(\Lambda J)^{n-k}\,M\,(\Lambda J^\top)^{k-1} \rangle_{(v,\tilde{v})\sim\mathcal{N}\left[(\bar{v},\bar{\tilde{v}}),G\right]} \overset{1\le k\le n-1}{\Rightarrow} \left[\bar{R}\right]^{n-k-1}\bar{R}\left[\bar{R}^\top\right]^{k-2}\bar{\tilde{C}},$$

which produce (after differentiation w.r.t. $\tilde{C}(u,w)$)

$$\sum_{n=2}^\infty \sum_{k=1}^{n-1} R^{n-k}(t,s)\left(R^\top\right)^k(u,w)\tilde{C}(t,w) = \sum_{n=1}^\infty \sum_{k=1}^{n} R^{n-k+1}(t,s)\left(R^\top\right)^k(u,w)\tilde{C}(t,w)$$

$$= \sum_{l=1}^\infty \sum_{k=1}^\infty R^l(t,s)\left(R^\top\right)^k(u,w)\tilde{C}(t,w)$$

$$= \left(K^*\right)^\top(u-w)\tilde{C}(w,t)K^*(t-s)\,.$$

Putting everything together this yields:

$$\langle \xi_i(t)\xi_j(s) \rangle = \delta_{ij}\left[2\beta^{-1}\delta(t-s) + P\eta_i\left\{K * \bar{C} * K^\top\right\}(t,s)\right]$$

$$+ P\sigma^2\eta_i\delta_{ij}\left[1 + \int K^*(t-u)\,\mathrm{d}u + \int K^{*\top}(w-s)\,\mathrm{d}w + \int K^*(t-u)\,\mathrm{d}u \int K^{*\top}(w-s)\,\mathrm{d}w\right]$$

$$= \delta_{ij}\left[2\beta^{-1}\delta(t-s) + P\eta_i\left\{K * \bar{C} * K^\top\right\}(t,s)\right]$$

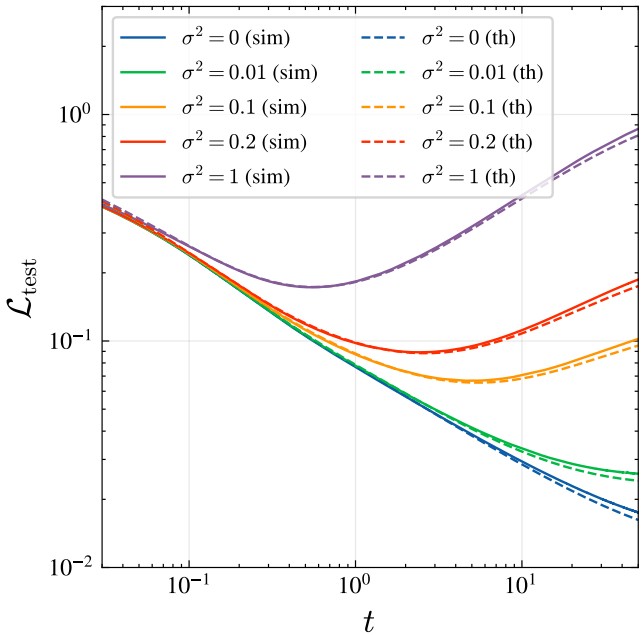

*Figure 6.* Test error for different strengths $\sigma^2$ of the label noise with $\sigma^2 = 0.0$ (blue), $\sigma^2 = 0.01$ (green), $\sigma^2 = 0.1$ (orange), $\sigma^2 = 0.2$ (red) and $\sigma^2 = 1.0$ (purple). The solid curves show the simulation, the dashed curves the theory. The other parameters are $g\beta = 10^3$, $P = N = 100$, $\Lambda_{ij} = i^{-3/2}\delta_{ij}$ and $\beta = 10^3$. The time step used for the simulation is $\mathrm{d}t = 10^{-4}$, for the theory $\mathrm{d}t = 10^{-2}$. The disorder average in the simulation is taken over $10^5$ different realizations of training data.

$$+ P\sigma^2\eta_i\delta_{ij}\left[1 + \int K^*(t-u)\mathrm{d}u\right]\left[1 + \int K^*(s-u)\mathrm{d}u\right]^\top$$

$$= \delta_{ij}\left[2\beta^{-1}\delta(t-s) + P\eta_i\left\{K * \bar{C} * K^\top\right\}(t,s)\right]$$

$$+ P\sigma^2\eta_i\delta_{ij}\left[\int K(t-u)\mathrm{d}u\right]\left[\int K(s-u)\mathrm{d}u\right]^\top \quad .$$

## I. Stationary Limit

We want to show, that our approach is consistent with previous results from the literature, where as a reference we show the equivalence to eq. 4 of (Canatar et al., 2021) who consider the static case in the noiseless limit $\beta \to \infty$. First, we note that $\frac{1}{g\beta\hat{K}(0)}$ can be identified with their $\kappa$ as they fulfill the same self-consistency equation, which can be seen from rewritten for $\omega = 0$ as

$$\kappa := \frac{1}{g\beta\hat{K}(0)} = \frac{1}{g\beta}\left(1 - \sum_i \eta_i\hat{G}_i(0)\right)$$

$$= \frac{1}{g\beta}\left(1 + \sum_i \frac{\eta_i}{\frac{1}{g\beta} + P\eta_i\hat{K}_i(0)}\right)$$

$$= \frac{1}{g\beta} + \frac{1}{g\beta\hat{K}(0)}\sum_i \frac{\eta_i}{\frac{1}{g\beta\hat{K}(0)} + P\eta_i}$$

$$= \frac{1}{g\beta} + \kappa\sum_i \frac{\eta_i}{\kappa + P\eta_i},$$

where $1/g\beta$ is equivalent to their $\lambda$ and we used the Fourier transform of (23) using (24) in the first and (26) in the second step. From this equation we can see the useful relation

$$\hat{G}_i(0) \equiv g\beta \frac{\kappa}{\kappa + P\eta_i} \quad .$$

First note that the test error in the stationary case also decomposes into the bias and variance part

$$\begin{aligned}
\mathcal{L}_{\text{test}}(t \to \infty) &= \frac{1}{2}\bar{C}(t = s \to \infty) \\
&= \frac{1}{2}\sum_i \eta_i \langle v_i(t \to \infty)\rangle^2 + \frac{1}{2}\sum_i \eta_i \langle \delta v_i(t \to \infty)\rangle^2 \,.
\end{aligned}$$

The bias part in the long time limit can be evaluated using (28) by writing it as

$$\mathcal{L}_{\text{bias}}(t \to \infty) = \frac{1}{2}\sum_i \eta_i \frac{\hat{G}_i^2(0)}{(g\beta)^2}\bar{w}_i^2 \equiv \frac{1}{2}\kappa^2 \sum_i \eta_i \left(\frac{\bar{w}_i}{\kappa + P\eta_i}\right)^2 \,, \tag{79}$$

in accordance to supplementary eq. (76) of (Canatar et al., 2021).

To evaluate the variance part, we use that for $\langle \delta v(t)\delta v(s)\rangle$ becoming stationary in both $t$ and $s$, we have from (30)

$$\frac{1}{2}\langle \delta v_i(t \to \infty)\delta v_i(s \to \infty)\rangle = P\eta_i \hat{G}_i(0)\,\hat{K}(0)\,\mathcal{L}_{\text{test}}(t \to \infty)\,\hat{K}^\top(0)\,\hat{G}_i^\top(0)\,,$$

so that we get

$$\begin{aligned}
\mathcal{L}_{\text{test}}(t \to \infty) &= \mathcal{L}_{\text{bias}}(t \to \infty) + P\sum_i \eta_i^2 \hat{G}_i^2(0)\hat{K}^2(0)\mathcal{L}_{\text{test}}(t \to \infty) \tag{80} \\
&=: \mathcal{L}_{\text{bias}}(t \to \infty) + \gamma\,\mathcal{L}_{\text{test}}(t \to \infty) \quad ,
\end{aligned}$$

where we introduced a new parameter $\gamma$ in the same manner as in (Canatar et al., 2021). This can be seen by looking at

$$P\sum_i \eta_i^2 \hat{G}_i^2(0)\hat{K}^2(0) = P\sum_i \eta_i^2 \left(g\beta\hat{K}(0)\right)^2 \left(\frac{\kappa}{\kappa + P\eta_i}\right)^2 = \sum_i \frac{P\eta_i^2}{(\kappa + P\eta_i)^2}$$

which is the $\gamma$ defined there.

Plugging (79) into (80) and solving for $\mathcal{L}_{\text{test}}(t \to \infty)$ we get

$$\mathcal{L}_{\text{test},\infty} = \frac{1}{1-\gamma}\sum_i \frac{\kappa^2 \eta_i \bar{w}_i^2}{(\kappa + P\eta_i)^2}\,,$$

which agrees to the result from (Canatar et al., 2021).

## J. Extension to committee machines

The framework derived can be extended also to non-linear settings and we want to introduce a possible extension here. Consider the following teacher-student setup (Krogh & Hertz, 1991) with

$$f_\mu := f(w, x_\mu) = \phi(w^\top x_\mu)\,, \tag{81}$$

$$y_\mu := \phi(\bar{w}^\top x_\mu)\,, \tag{82}$$

with data $x_\mu$ and $\mu$ indexes the training points. The labels $y_\mu$ are generated by the teacher weights $\bar{w} \in \mathbb{R}^d$ and $w \in \mathbb{R}^d$ are the student weights to be learned. The train set for this regression task is given by $\mathcal{D} = \{(x_\mu, y_\mu)\}_{\mu=1,...,P}$ and $(x_*, y_*) \notin \mathcal{D}$

is a test point. We assume $x_{\mu i}, x_{*i} \overset{\text{i.i.d.}}{\sim} \mathcal{N}(0, d^{-1})$ to be drawn from the same distribution. We solve the regression task on the teacher-student setup with a squared loss

$$H(w, \mathcal{D}) := \frac{1}{2} \sum_{\mu=1}^{P} \left( y_\mu - f(w, x_\mu) \right)^2 + \frac{1}{2g\beta} \|w\|^2. \tag{83}$$

Training the system with stochastic Langevin gradient descent yields

$$\frac{\partial}{\partial t} w_i(t) = -\frac{\partial}{\partial w_i} H(w, \mathcal{D}) + \zeta_i(t)$$

$$= \sum_{\mu=1}^{P} \left( \phi(\bar{h}_\mu) - \phi(h_\mu) \right) \phi'(h_\mu) \, x_{\mu i} - \frac{1}{g\beta} w_i + \zeta_i(t) \,,$$

$$\langle \zeta_i(t) \zeta_j(s) \rangle = \frac{2}{\beta} \delta_{ij} \, \delta(t - s) \,,$$

so that the stationary distribution is given by $\exp\left( -\beta H(w, \mathcal{D}) \right)$. This equation of motion can be seen as an equation of motion on the level of the pre-activations $h_\nu$ by defining $h_\mu := w^\top x_\mu = \sum_{i=1}^{d} w_i x_{\mu i}$ as well as $\bar{h}_\mu := \bar{w}^\top x_\mu$ and $\zeta_\nu(t) := \sum_{i=1}^{d} x_{\nu i} \zeta_i(t)$ to obtain

$$\frac{\partial}{\partial t} h_\nu = \sum_{\mu=1}^{P} \left( \phi(\bar{h}_\mu) - \phi(h_\mu) \right) \phi'(h_\mu) \sum_{i=1}^{d} x_{\mu i} x_{\nu i} - \frac{1}{g\beta} h_\nu + \zeta_\nu(t) \,.$$

Introducing the shorthands

$$C_{\mu\nu}^{(xx)} := \sum_{i=1}^{d} x_{\mu i} \, x_{\nu i} \,,$$

$$\delta\phi_\mu(h_\mu) := \phi(h_\mu) - \phi(\bar{h}_\mu) \,,$$

$$\Phi(h_\mu \,; \bar{h}_\mu) := \delta\phi_\mu(h_\mu; \bar{h}_\mu) \phi'(h_\mu) \,,$$

the equation of motion takes the form

$$\frac{\partial}{\partial t} h_\nu = -\sum_{\mu=1}^{P} C_{\nu\mu}^{(xx)} \, \Phi(h_\mu \,, \bar{h}_\mu) - \frac{1}{g\beta} h_\nu + \zeta_\nu(t) \,, \tag{84}$$

which corresponds to the following moment generating functional

$$Z = \int \mathcal{D}h \int \mathcal{D}\tilde{h} \, \exp \Big( \int dt \sum_{\nu=1}^{P} \tilde{h}_\nu(t) \left[ \left( \partial_t + \frac{1}{g\beta} \right) h_\nu(t) + \sum_{\mu=1}^{P} C_{\nu\mu}^{(xx)} \, \Phi_\mu(t) \right]$$

$$+ \frac{1}{\beta} \sum_{\mu,\nu=1}^{P} \tilde{h}_\mu(t) \, C_{\mu\nu}^{(xx)} \, \tilde{h}_\nu(t) \Big) \,.$$

The disorder average over $C_{\mu\nu}^{(xx)}$, which again appears linear in the exponent, can be treated in the same way as in the main text and is left for future work. Note, though, that within the matrix $C^{(xx)}$ the synapse index has been contracted, while in the linear case considered in the main text, the contraction is over the sample index. Still, both matrices are Wishart distributed and hence allow for the same treatment.

