# OpenReview forum: "Dynamics of neural scaling laws in random feature regression with powerlaw-distributed kernel eigenvalues"
_ICML.cc/2026/Conference — ICML 2026 regular_

### Official Review · Reviewer_a7iD · 2026-03-03

**Soundness:** 3
**Presentation:** 3
**Significance:** 2
**Originality:** 3
**Overall Recommendation:** 5
**Confidence:** 3

**Summary:**

The paper investigates kernel ridge regression for data with power-law spectra. Using Dynamical Mean-Field Theory (DMFT), MSRDJ formalism, and variational Gaussian approximation, the authors introduce a temporal memory kernel (without auxiliary field) that simplifies the stochastic Langevin dynamics of the system. The theory successfully decouples the collective dynamics into independent eigenmodes, where a mean-field coupling determines the discrepancy between each mode and its true teacher value. Finally, the authors provide a theoretical justification for early stopping by demonstrating how it prevents the variance contribution of the generalization error from dominating over time.

**Compliance With Llm Reviewing Policy:**

Affirmed.

**Final Justification:**

The authors have clearly articulated the paper's novelty and proposed an effective way to convey it, justifying a score increase.

**Key Questions For Authors:**

Given that prior DMFT works also push spatial randomness into the time domain, could the authors elaborate on why their specific method of avoiding auxiliary fields yields a simpler physical or intuitive interpretation of the memory kernel?

**Limitations:**

yes

**Strengths And Weaknesses:**

## Strengths
 1. **Simplification of DMFT dynamics:** The memory kernel offers a more compact description of kernel dynamics compared to prior DMFT works. The decoupling of the eigenbasis dynamics and the avoidance of auxiliary fields by averaging over the kernel directly represent a meaningful technical contribution. The resulting terms are relatively more intuitive and have the potential to be extended to more complicated setups, such as non-linear networks, in the future.

 2. **Strong Empirical Validation:** Unlike many modern ML papers that rely on weak evidence, this work provides a rigorous quantitative comparison between theoretical predictions and numerical simulations. Additionally, the simulations are extensive, including an average over $10^5$ data realizations in Fig. 2 to validate the test error dynamics.

## Weaknesses
 1. **Incremental Contribution:** As the authors acknowledge, the static limit ($t \rightarrow \infty$) and the effects of power-law spectra on generalization have been investigated extensively, such as by Canatar et al. (2021) . The DMFT of learning dynamics has also been studied recently (e.g., Bordelon et al., 2024; Bordelon et al., 2025). While the specific Langevin techniques and assumptions differ, the core qualitative findings on dynamics are not entirely novel.

 2. **Presentation:** As mentioned in the strengths, the technical simplification by averaging the kernel itself without using auxiliary field is novel. However, as the results are relatively incremental, I believe the paper would have benefited from explaining the physical intuition behind it or why it is intuitively simpler. For instance, at the end of Section 4.3, the authors briefly state that slow modes become more important over time due to their "long range Green's functions" and that L2 regularization restrains the variance by enforcing decay on these functions. However, this is difficult to understand before reading the definition of memory kernel in the appendices. Personally, this is a critical weakness because I highly value such intuitions, and the authors had an extra page they did not use, as the main text ends on page 7. Perhaps bringing some results in Appendices such as memory kernel as geometric series of the system's perturbation response in Appendix B,C, and D better sets up the scene for the discussion later. More elaboration in this area would have increased my score.

 3. **Minor polishing issues**:
     * Running title is missing
     * Reference that need formatting:
        Bordelon, B., Atanasov, A., and Pehlevan, C. How feature learning can improve neural scaling laws **¡sup¿*¡/sup¿.** Journal of Statistical Mechanics Theory and Experiment, 2025. doi: 10.1088/1742-5468/adefb1. 2
     * Inconsistent formatting for three authors paper: e.g. Bordelon, Atanasov, and Pehlevan (2025) (page 2)
     * Repeated referencce for Jacot, A., Gabriel, F., and Hongler, C (same paper cited as 2018 and 2020)
     * Notation definitions not promptly shown: Langevin noise $\zeta$ is first used in eq 4 (page 2), but only mentioned at page 5.
     * Some reference parenthesis formatting issue. For example (see e.g. (Naveh et al., 2020)) in page 2 should be (see e.g. Naveh et al., 2020)

---

> ### Author Rebuttal · Authors · 2026-03-31
>
> # Incremental contribution
> In a revision we will explain how our approach relates to earlier work. Please also see reply "Embedding into literature" to referee **NCZD**. We will also explain on a technical level how our approach differs from Bordelon et al. 2025. Please see details in reply "Difference of approach on technical level" to **NCZD**.
>
> In the revision we will add
>
> 1. A derivation for the scaling law of the Pareto frontier observed in Figure 1a of Kaplan et al. 2020. Our derivation shows how it arises from an interplay of the declining bias part of the
> loss and an increase of the variance part (see  https://figshare.com/s/3fc3e9dcf813d874c8d8, fig. 1 for a preliminary figure)
>
> 2. An Appendix explaining how our approach may be extended to study non-linear regression in committee machines (see also reply "Extension beyond linear setting" to **CeD1**.
>
> 3. An extension of our approach to include label noise (cf. fig. 2), so that our dynamical formulation now covers all common setups; this latter addition corresponds in the dynamics, to the step done from Bordelon et al. 2020 to Cui et al. 2021 in the static case.
>
> # Presentation
> We agree that intuitive interpretations are most helpful and in a revision will better explain how the modes with slow timescales will become important as those "collecting" noise and ultimately limiting the  generalization error.
> We will also add intuition with regard to the meaning of the memory term (please see our reply "Technical differences of approaches and interpretability" to **NCZD**).
>
> # Minor polishing issues
>
> We thank the referee for pointing us to these minor polishing issues and will carefully correct them.
>
> # Questions
>
> Q: Given that prior DMFT works also push spatial randomness into the time domain, could the authors elaborate on why their specific method of avoiding auxiliary fields yields a simpler physical or intuitive
> interpretation of the memory kernel?
>
> A: In our approach we have one random projection less compared to Bordelon et al. 2025 (they have an additional mapping into a lower dimensional feature space). As a result, we only need $2$ order parameters (one response function and one autocorrelation function), rather than $8$ order parameters. Compared to classical approaches of dynamical spin glass theory (e.g., Sompolinsky \& Zippelius 1982; Crisanti et al. 1987), $2$ appears to be the minimal number of order parameters, since the symmetry of the matrix $\psi \psi^{\top}$ leads to Onsager-reaction terms (the non-local kernel $K$) and likewise to an effective Gaussian noise ($C$). The kernel $K$ intuitively appears because the frozen variability in the symmetric part of $\psi \psi^T$ causes fluctuations of weight $i$ influencing weight $j$ to return with the same sign to weight $i$ at a later time.
> Please also see reply "Technical differences of approaches and interpretability" to referee **NCZD**.

---

> > ### Author Rebuttal · Reviewer_a7iD · 2026-04-01
> >
> > I thank the authors for their detailed rebuttal. I believe that incorporating more intuition, expanding on the relationship to prior work, and further polishing the manuscript will make this a very valuable contribution.
> >
> > Although some results overlap with existing literature, the paper introduces a cohesive dynamical framework with clear technical and intuitive novelty. I am pleased to raise my score.

---

### Official Review · Reviewer_CeD1 · 2026-03-09

**Soundness:** 2
**Presentation:** 3
**Significance:** 3
**Originality:** 2
**Overall Recommendation:** 3
**Confidence:** 4

**Summary:**

This paper develops a dynamical mean‑field theory for the generalization dynamics of high‑dimensional linear or kernel regression with power‑law‑distributed kernel eigenvalues. By employing the MSRDJ formalism and a variational Gaussian approximation, the authors derive a self‑consistent effective description in terms of low‑dimensional order parameters, such as response and correlation functions. The resulting framework unifies several previously studied regimes, including Bayesian inference in Gaussian process regression, gradient flow with or without weight decay, and stochastic Langevin training. Within this framework, the paper analyzes the dynamics of the generalization error and its bias‑variance decomposition, providing a spectral‑dynamical interpretation of early stopping and the effect of $L_2$ regularization under noisy training conditions.

**Compliance With Llm Reviewing Policy:**

Affirmed.

**Final Justification:**

Despite the authors' substantial efforts, the additional argument provided in the rebuttal still relies largely on heuristic discussion, and I remain unconvinced that the key claim regarding this point has been fully substantiated. Moreover, this issue does not seem easy to resolve completely within the rebuttal stage. Therefore, I will maintain my original score.

**Key Questions For Authors:**

See Weaknesses

**Limitations:**

Yes

**Strengths And Weaknesses:**

### Strengths

- The paper addresses a timely and important problem by studying generalization dynamics on data with power‑law spectral structure. Its attempt to connect several learning regimes, including Bayesian inference, gradient flow, and stochastic Langevin dynamics, within a common analytical framework is a meaningful conceptual contribution.

- The paper demonstrates a reasonably strong match between theory and simulation in the presented synthetic settings. In particular, Figures 2‑4 indicate that the dynamical mean‑field predictions capture the main qualitative and quantitative trends observed in numerical experiments, including the evolution of test error, its bias‑variance decomposition, and the effect of regularization.

- Another strength is the move beyond purely static or equilibrium analyses toward an explicitly time-dependent description. By deriving effective dynamics for response and correlation statistics, the paper provides an interpretable spectral-dynamical picture of the bias–variance trade-off and a mechanism-based explanation for why early stopping can help in noisy training.

### Weaknesses

- The paper’s title and introduction repeatedly invoke “neural scaling laws.” However, the actual object of study is restricted to linear regression / kernel regression with power-law eigenvalue spectra. Since realistic neural scaling behavior is often thought to depend, at least in part, on nonlinear feature learning, and the authors themselves acknowledge that the present framework is confined to the linearized setting, the scope of the claims around explaining neural scaling laws appears broader than what is directly established in the paper.

- The use of a variational Gaussian approximation is reasonable as a tractable and physically motivated ansatz, but the claim that it becomes exact as $N\to\infty$ because of the central limit theorem appears insufficiently justified. The effective process remains self-consistently coupled through collective response/correlation terms and temporal memory kernels, so Gaussianity does not seem to follow automatically from a naive CLT argument. Establishing exactness would likely require a more explicit dynamic mean-field/cavity argument, or at least a clearer derivation of the limiting Gaussian process and the assumptions under which the variational ansatz is asymptotically exact.

- The modeling assumptions are strong and fairly idealized: the features are Gaussian and independent across samples in the kernel eigenbasis, and the population covariance is diagonal by construction. While these assumptions make the analysis tractable, they also leave open how robust the resulting phenomena, such as effective colored noise, mode‑dependent slowdown, and the proposed spectral interpretation of early stopping, are beyond this stylized teacher‑student setting. The paper currently offers limited discussion of this issue.

- Many ingredients of the paper are not individually new: power-law kernel spectra, kernel regression, the Bayesian/GP correspondence, DMFT/MSRDJ methods, and bias-variance interpretations of early stopping all have clear prior foundations. The originality here lies primarily in combining these ingredients into a unified dynamical framework. That may still be valuable, but the paper would benefit from a sharper articulation of what is genuinely new relative to the closest prior work; otherwise, the contribution may come across as more integrative than fundamentally novel.

---

> ### Author Rebuttal · Authors · 2026-03-31
>
> # Extension beyond linear setting
>
> In the revision we will present the extraction of the power law of the Pareto frontier (as empirically described in Kaplan et al. 2020, Figure 1 a)
> from an approximation of our effective equation of motion. Please see our reply "Prediction of scaling laws" to referee **NCZD** and also the supplementary figure 1 (https://figshare.com/s/3fc3e9dcf813d874c8d8).
>
> Moreover, in the revision we will add an appendix showing that the approach is also applicable to learning in non-linear networks. Concretely, we will show that a teacher-student setup of committee machines with student $f_{\mu} = \sum_i \phi(w_i^\mathrm{T} x_{\mu,i})$ and teacher $y_{\mu} = \sum_i \phi( \bar{w}^T_i x_{\mu,i})$ may allow a similar analysis. We would like to leave the complete analysis for future work, but share the expectation by the referee that this would be an important further step; for example to investigate the learning dynamics in systems with computational to statistical gaps, as they were demonstrated in Aubin et al. 2019 [https://arxiv.org/abs/1806.05451] in the static case.
>
> # Justification of VGA
>
> We agree that our argument was too brief to be appreciated and we will extend it in a revision. We would here like to stay on the level
> of a conjecture as usual in a statistical physics approach. In brief, the argument starts with the exact expression Eq. (35). The expansion in Eqs. (36)-(38) shows that concentrating quantities appear, $R, \tilde{C}, C$, due to the extensive sums $\sum_{k=1}^{N}$. One notes that
>
> 1.) the same order parameters re-appear also in the higher order terms of the expansion and
>
> 2.) that after identification of each of these sums with the corresponding order parameter, the remaining action is Gaussian,
>
> 3.) it factorizes over weight indices $i$; this factorization (vanishing correlations) in turn, is required for the order parameters to concentrate to their mean by CLT, and
>
> 4.) terms dropped in the expansion are suppressed by $\mathcal{O}(N^{-1})$ or more.
>
> # Idealized Assumptions
>
> We will add an extended discussion of the assumptions in a revision. The assumption of Gaussian features appears widespread and, if the
> data dimension scales in proportion to the model dimension, is often referred to as "Gaussian equivalence" (Hastie et al., 2022; Hu and Lu, 2022b; Montanari and Saeed, 2022; Misiakiewicz and Saeed, 2024); see also Atanasov et al. for a comprehensive overview [https://arxiv.org/pdf/2405.00592]. This assumption, even outside the proper limit, is often found to empirically agree well on real data; see, e.g., [https://arxiv.org/pdf/2510.25553] Figure 12 for CIFAR10.
>
> We would also like to preempt or correct a potential misunderstanding: Our model does not assume a diagonal empirical covariance matrix $\Psi \Psi^T$; rather we assume the features $\Psi$ to be uncorrelated Gaussian (by Gaussian equivalence). This independence translates into $\langle \Psi \Psi^{\top}\rangle$ being diagonal only on expectation; any single draw of that (non-white) Wishart matrix, instead, will typically not be diagonal (please see suppemental figure 3).
>
> We agree to the referee that our discussion of the other approaches has been too short. In particular, highlighting the seminal steps
> made by previous works. We will amend our discussion of related works; please see our reply to **NCZD** "Embedding into the literature" and also "Technical differences of approaches and interpretability".

---

> > ### Author Rebuttal · Reviewer_CeD1 · 2026-04-02
> >
> > I appreciate the authors’ efforts in addressing the reviewers’ comments.
> >
> > The rebuttal clarifies some of the authors’ intentions, but my core concerns have not been substantively resolved. On several key issues, the authors mainly acknowledge the limitations of the current version and explain the motivation behind their current treatment as well as their plans for revision, rather than actually resolving these issues at the rebuttal stage. For example, regarding the justification of VGA, I understand why the authors chose this approach, but the response still remains largely at a heuristic level and is not sufficient to substantially remove my concerns on this point.
> >
> > Therefore, although I appreciate that the authors’ response is serious and helpful, I do not find it sufficient to change my overall assessment of the current submission, and I therefore keep my original score unchanged.

---

> > > ### Author Response · Authors · 2026-04-06
> > >
> > > We thank the referee for coming back with additional requests for clarification. In the previous reply we have
> > >
> > > - 1. Clarified that the Gaussian approximation of features is backed up by literature and goes under the name Gaussian equivalence principle (Hastie et al., 2022; Hu and Lu, 2022b; Montanari and Saeed, 2022; Misiakiewicz and Saeed, 2024)
> > >
> > > - 2. Made explicit predictions for the Pareto frontier of neural scaling laws (see our reply to NCZD and additional material https://figshare.com/s/3fc3e9dcf813d874c8d8)
> > >
> > > - 3. Explained that the memory terms, which cause a slowdown of the dynamics, will in a qualitatively similar manner also appear in non-linear learning systems, such as committee machines.
> > >
> > > - 4. Explained which steps occur in the VGA and how different terms behave in the large N limit, justifying why the process indeed becomes Gaussian in the limit, so that the VGA becomes exact.
> > >
> > > The referee now asks for more technical details for point 4., which we are of course happy to provide. In the following we want to elaborate on our previous answer and sketch an improved derivation of the $N\to\infty$ limit, showing that all non-Gaussian terms indeed vanish in the limit, so that the VGA becomes exact.
> > >
> > >
> > > ## Exactness of the VGA in the N→∞ Limit
> > >
> > > ### Concentration of order parameters
> > >
> > > In the limit N→∞, the macroscopic correlation C and response R concentrate to their mean values by the CLT, once (ṽ_k, v_k) become statistically independent across modes k. The power-law spectrum $η_k = k^{-(1+α)}$ with $α > 0$ yields a convergent sum $Σ_k η_k → α⁻¹$, which majorizes $C$ and $R$, guaranteeing both exist in the limit.
> > >
> > > ### Dropped terms vanish or are sub-leading
> > >
> > > We show that the terms dropped either vanish identically or are sub-leading in N→∞. Consider n=2:
> > >
> > > $$tr[ΛJΛJ] = Σ_{ik} η_i η_k ∫∫ ṽ_i(t) v_k(t) ṽ_k(s) v_i(s) dtds.$$
> > >
> > > Evaluating this perturbatively, two qualitatively distinct contractions arise.
> > >
> > > **First contraction — equal-time response.** Contracting the pair ṽ_k(s) v_i(s) at time s yields
> > >
> > > $$Σ_{ik} η_i η_k ∫ ṽ_i(t) v_k(t)dt ∫ <ṽ_k(s) v_i(s)>ds,$$
> > >
> > > which would couple modes i and k through the time-integrated response ∫<ṽ_k(s) v_i(s)> ds. However, <ṽ_k(s) v_i(s)> is an equal-time response function, and in the Ito convention the retarded propagator satisfies G(t,s)=0 for t ≤ s. Therefore this contraction **vanishes identically** at any N.
> > >
> > > **Second contraction — causal self-coupling.** Contracting v_k(t) ṽ_k(s) at times t > s yields
> > >
> > > $$Σ_i η_i ∫∫ ṽ_i(t) [Σ_k η_k <v_k(t) ṽ_k(s)>] v_i(s)dtds.$$
> > >
> > > This is non-zero because the causal response <v_k(t) ṽ_k(s)> ≠ 0 for t > s. Crucially, the outer fields ṽ_i(t) and v_i(s) carry the **same** mode index i. This term mediates only a self-coupling of mode i and generates no inter-mode correlations. It is precisely the contribution retained by the VGA.
> > >
> > > ### Extension to all orders n > 2
> > >
> > > For higher powers, products $ṽ_i(r) v_j(r) ṽ_j(s) v_k(s) ··· ṽ_l(t) v_i(t)$ arise. Any cross-mode contraction $<ṽ_i(r) v_k(s)> ≠ 0$ requires r < s by causality, but the remaining contractions produce $<v_j(r) ṽ_j(s)>$ requiring r > s — a contradiction. Hence all cross-mode response pairings vanish at every order.
> > >
> > > ### Cross-mode correlations are O(1/N) sub-leading
> > >
> > > Pairing two v-fields generates correlated noise ∝ <v_k(t) v_i(s)> between modes. First, we note that vanishing cross-correlations are a consistent solution (the driving noise ζ_i is uncorrelated across modes). But each such term only contributes O(1), whereas the self-coupling tr[ΛJ ΛJ^T] sums N variance contributions to O(N). Cross-correlations are in any case thus suppressed by 1/N.
> > >
> > > ### Terms with $\tilde{C}^2$ or $\tilde{\bar{v}}$ vanish by Ito
> > >
> > > Higher-order terms like $<tr JJ^T JJ^T> ~ \tilde{C}^2$ vanish: one $\tilde{C}^2$ generates effective Gaussian noise, but the second requires <ṽ ṽ> ≡ 0 by causality. In the VGA this is enforced by setting $\tilde{C}^2$= 0 after differentiating the equation of state. Analogously, terms with ≥2 factors of $\tilde{\bar{v}}$ vanish because <ṽ> ≡ 0 after differentiation.
> > >
> > > ### Conclusion
> > >
> > > All leading-order terms in N→∞ involve only the concentrating quantities C or R. The sub-leading terms — precisely those coupling modes i ≠ k — are dropped, leaving a **Gaussian action** with independent modes conditioned on C and R. This establishes that the VGA is **asymptotically exact as N→∞**.

---

### Official Review · Reviewer_NCZD · 2026-03-13

**Soundness:** 3
**Presentation:** 2
**Significance:** 2
**Originality:** 2
**Overall Recommendation:** 4
**Confidence:** 4

**Summary:**

The paper studies the learning dynamics of random-feature regression under Langevin training when the kernel spectrum follows a power-law decay. Using a MSRDJ formulation, under a variational Gaussian approximation, the authors derive effective dynamical mean-field equations describing the evolution of individual kernel eigenmodes. The framework yields a bias–variance decomposition of the time-dependent generalization error and provides an interpretation of early stopping through mode-dependent learning timescales. The analytical findings contributes to the understanding of the learning dynamics in high-dimensional models with structured spectra.

**Compliance With Llm Reviewing Policy:**

Affirmed.

**Final Justification:**

The authors' rebuttal addressed my points by promising additional important results and revisions, and by clarifying their contribution within the literature. Therefore, I am increasing my score.

**Key Questions For Authors:**

1. Could the author extended the discussion in the manuscript around line 357, on how their technical approach offers more interpretable results with respect with previous works' findings? A more explicit comparison would enhance clarity and the significance of this work.
2. Are the results consistent with the one derived in the limit $t\to\infty$ in the existing literature?
3. Concerning Weaknesses 1 and 2, could the analytical framework developed in this work be used to derive more explicit quantitative predictions, such as scaling laws for the generalization error? If not straightforward, what would be the main obstacle?

**Limitations:**

Yes

**Strengths And Weaknesses:**

### Strengths

The paper provides an interesting perspective on the problem of random feature regression, combining a finite-time analysis to the effects of regularization, through the interpretable lens of a statistical physics formulation. The result on dynamics of individual eigenmodes offers an intuitive explanation on timescale separations. Finally, numerical experiments are consistent with the theoretical findings.

### Weaknesses
1.  Although stated as one of the motivations, and in the title, the paper does not provides quantitative scaling laws for the test error, depending on the spectral structure, leaving the dependence on the decay exponent $\gamma$ mostly implicit in their result and not quantitavely interpretable.
2. Similarly, the motivation around the benefits of early stopping is mostly qualitative, with no predictive results on suitable stopping time that depends on the problems' parameters.
3. The setting is limited to the linear design with matching features between teacher and student.
4. The paper could benefit from a better positioning in the existing literature (see Questions 1-2). Further, there are missing citations to the scaling laws literature in kernel and random feature regression under power-law spectrum assumptions, closely related to the setting of this work. Few examples:
- Bordelon et al., "Spectrum Dependent Learning Curves in Kernel Regression and Wide Neural
Networks", 2020
- Spigler et al., "Asymptotic learning curves of kernel methods: empirical data versus teacher–student paradigm", 2020
- Cui et al., " Generalization error rates in kernel regression: the crossover from the noiseless to noisy regime.", 2022
-  Paquette et al., "4+3 phases of computeoptimal neural scaling laws", 2024
- Defilippis et al., "Dimension-free deterministic equivalents and scaling laws for random feature regression", 2024
- Atanasov et al. "Scaling and renormalization in high-dimensional regression", 2024

---

> ### Author Rebuttal · Authors · 2026-03-31
>
> # Prediction of scaling laws
>
> In the revision we will extend our analysis of the scaling law Pareto frontier (as empirically described in Kaplan et al. 2020, Figure 1 a) by characterizing the bias-variance tradeoff.
> Approximating $K\simeq\delta$, we derive a lower bound for the weight decay and obtain universal scaling laws, identifying the
> training noise contribution L\_beta as dominant to the variance.
> In the supplementary fig. 1 ( https://figshare.com/s/3fc3e9dcf813d874c8d8) we show the arising power-laws.
> The optimal stopping time $t^\star$ follows from $\frac{dL_{test}}{dt}\simeq 0$, yielding the universal relation $2t^\star = \alpha \beta/(1+\alpha)$ between $t^\star$, batch size $P$, and spectral exponent $\alpha$. Prior to $t^\star$, the test error is dominated by $L_{bias}\sim c_1(Pt)^{-\alpha/(1+\alpha)}+c_2 N^{-\alpha}$ (fig. 1 c)); compute-optimal training is therefore constrained to $t<t^\star$.
>
> Surprisingly, while the optimal test error at $t^\star$ depends on the gradient noise $\beta$, the loss evaluated at the optimal compute
> budget $f^\star=6PNt^\star$ [https://arxiv.org/pdf/2405.15074] is independent of both $\beta$ and $P$, scaling as $L_{test}(f^\star)\sim(f^\star/N)^{-\alpha/(1+\alpha)}$ for $N\gg 1$. (fig.1 b))
>
>
> # Embedding into the literature
>
> We agree these closely related works deserve more discussion and will update the related works section accordingly.
> We begin with static approaches that do not capture learning dynamics, noting where our stationary solutions agree:
>
> Bordelon et al. 2020 compute the generalization error of kernel ridge regression in a student-teacher setting with power-law data. Our
> $t\to\infty$ limit recovers their results when Langevin noise vanishes ($\beta\to\infty$); for $\beta<\infty$, our stationary point in addition
> yields the Bayesian posterior. Beyond statics, our theory predicts the full time evolution at finite training time.
> Cui et al. 2022 extend Bordelon et al. by adding label noise which we will include in our revision, reproducing their $t\to\infty$ results (cf. fig. 2).
> Spigler et al. 2020 compare student-teacher kernel regression to soft SVM classification and derive generalization error exponents from smoothness and dimensionality assumptions. Defilippis et al. 2024 study static random feature regression without specific scaling
> assumptions on sample or system size.
> Atanasov et al. 2024 derive deterministic equivalents for static random feature regression via the S-transform, compactly charting scaling law exponents across many settings.
>
> We now turn to dynamical approaches:
> Paquette et al. 2025 study the compute-optimal Pareto frontier in random feature regression trained by SGD with finite batch size and
> power-law features, deriving a deterministic Volterra equation via RMT and identifying multiple scaling phases. Like
> Bordelon et al. 2025, they use a down-projected student-teacher setup. We instead study Langevin gradient flow with identical student
> and teacher features, which, beyond feature regression, naturally connects to the Bayesian posterior.
>
>
> # Technical differences of approaches and interpretability
>
> In a revision, we will better explain the technical relation to previous work.
>
> Our approach is most closely related to Bordelon et al. 2025, who derive a dynamical mean-field theory in a teacher-student setting
> where the student learns on a sub-sampled feature set. Their theory requires eight self-consistently coupled order parameters (four
> correlation and four response functions), whereas ours requires only two: one autocorrelation function $C$ and one response function
> $R$, which is the minimal number expected also from dynamical spin glass theory [Sompolinsky \& Zippelius 1982; Crisanti et al. 1987].
> The temporally non-local response function in our theory, as in spin glasses, arises from the symmetry of the feature-feature matrix and can be understood as an Onsager reaction term: since weights interact symmetrically via $\psi\psi^T$, a fluctuation from weight $i$ influencing weight $j$ at time $t$ returns to weight $i$ at a later time $t'$ with identical sign. Neglecting this term and setting $K(t,s)\simeq\delta(t-s)$ corresponds to replacing the non-diagonal matrix $\psi\psi^T$ by its diagonal average $<\psi\psi^T>$ (cf. fig. 3).
> Finally, unlike Bordelon et al. 2025, our framework includes temporal training noise, which we find essential for predicting the Pareto frontier (see "Prediction of scaling laws" above).
>
>
> # Consistency with static approach
>
> Yes, our $t\to\infty$, $\beta\to\infty$ limit reduces to Bordelon et al. 2020: the Langevin process (Eqs. (4,5)) samples from $w\sim\exp(-\beta H)$, which for $\beta\to\infty$ minimizes $H$, recovering their Eq. (1) from our Eq. (3). In the revision we will add an appendix detailing the comparison of the static bias and variance (obtainable from our Eq. (26)) to their results, and verify numerical agreement.

---

> > ### Author Rebuttal · Reviewer_NCZD · 2026-04-01
> >
> > I thank the authors for their rebuttal and supplementary figures. My concerns regarding the absence of quantitative scaling laws and stopping time predictions have been addressed by the preliminary results provided in the reply. The authors are committed to extending this analysis in the revised version. Additionally, they have clarified their positioning within the scaling laws literature. I believe that implementing these revisions will increase the work's significance and overall presentation.

---

> > > ### Author Response · Authors · 2026-04-06
> > >
> > > We are glad to hear that we could resolve all concerns of the reviewer and want to thank them again for their insightful questions and suggestions that helped us to improve our manuscript. We hope the reviewer will consider to raise the score or otherwise come back with additional requests for clarification.

---

### Decision · Program_Chairs · 2026-04-30

**Decision:**

Accept (regular)

**Comment:**

This paper studies learning dynamics in random feature / kernel regression with power-law spectra using a dynamical mean-field approach. Reviewers agreed that the paper is technically solid, the theoretical predictions match simulations well, and the framework gives an interesting dynamical interpretation of generalization, early stopping, and spectral effects.

The main concerns were about positioning and scope: the title and motivation invoke “neural scaling laws” somewhat more broadly than what is directly established, and one reviewer remained unconvinced by the level of rigor supporting the asymptotic exactness of the variational Gaussian approximation. That said, the rebuttal addressed many concerns, clarified the relation to prior work, and strengthened the presentation of the contribution.

Overall, I view this as a worthwhile theory contribution with clear technical merit, though with some limitations in scope and rigor that should be improved in the final version. I therefore recommend Weak Accept.